# A General Framework for Black-Box Attacks Under Cost Asymmetry

**Mahdi Salmani**
University of Southern California
salmanis@usc.edu

**Alireza Abdollahpoorrostam**
EPFL
alireza.abdollahpoorrostam@epfl.ch

**Seyed-Mohsen Moosavi-Dezfooli**
Apple
smoosavi@apple.com

## Abstract

Traditional decision-based black-box adversarial attacks on image classifiers aim to generate adversarial examples by slightly modifying input images while keeping the number of queries low, where each query involves sending an input to the model and observing its output. Most existing methods assume that all queries have equal cost. However, in practice, queries may incur *asymmetric costs*; for example, in content moderation systems, certain output classes may trigger additional review, enforcement, or penalties, making them more costly than others. While prior work has considered such asymmetric cost settings, effective algorithms for this scenario remain underdeveloped. In this paper, we introduce asymmetric black-box attacks, a new family of decision-based attacks that generalize to the asymmetric query-cost setup. We develop new methods for boundary search and gradient estimation when crafting adversarial examples. Specifically, we propose *Asymmetric Search (AS)*, a more conservative alternative to binary search that reduces reliance on high-cost queries, and *Asymmetric Gradient Estimation (AGREST)*, which shifts the sampling distribution in Monte Carlo style gradient estimation to favor low-cost queries. We design efficient algorithms that reduce total attack cost by balancing different query types, in contrast to earlier methods such as *stealthy attacks* that focus only on limiting expensive (high-cost) queries. We perform both theoretical analysis and empirical evaluation on standard image classification benchmarks. Across various cost regimes, our method consistently achieves lower total query cost and smaller perturbations than existing approaches, reducing the perturbation norm by up to 40% in some settings. The code for *Asymmetric Attacks* is available at github.com/mahdisalmani/Asymmetric-Attacks.

## 1 Introduction

Decision-based adversarial attacks, first introduced by Brendel et al. (2017), generate adversarial examples in black-box settings by systematically querying a classifier and observing only its output decisions for perturbed inputs. The original Boundary Attack (Brendel et al., 2017) initially required over 100,000 queries to reliably identify minimal adversarial perturbations for large-scale datasets such as ImageNet (Deng et al., 2009). Subsequent works (Chen et al., 2020; Chen & Gu, 2020; Cheng et al., 2018; 2019; Liu et al., 2019b; Rahmati et al., 2020) significantly enhanced the efficiency of decision-based attacks by reducing the number of queries needed, achieving improvements of one to three orders of magnitude. These advancements have led to more practical and efficient frameworks for adversarial testing in limited-query settings.

While prior work (discussed in detail in App. A) has primarily assumed that all queries have equal cost and focused on minimizing the total number of queries, in many practical scenarios, queries can incur asymmetric costs depending on their nature. For instance, Not Safe for Work (NSFW) image detection models have become increasingly important, with major platforms such as Facebook (Facebook, 2024) and 𝕏 (formerly Twitter)(Twitter, 2024a) deploying automated mechanisms for identifying sensitive

content, alongside commercial APIs developed by Google(Google, 2024), Amazon (Amazon, 2024), and Microsoft (Microsoft, 2024). In these settings, submitting explicit or borderline explicit queries could trigger more severe consequences, such as account suspension or content flagging, compared to benign queries. As a result, minimizing only the total number of queries is insufficient; effective attack strategies must account for the asymmetric costs associated with different types of queries.

Debenedetti et al. (2024) introduced **stealthy attack** techniques to better handle asymmetric query costs. They empirically demonstrated that the standard binary search procedure for boundary point detection, mostly for projecting an adversarial point onto the decision boundary or for OPT-style gradient estimation (Cheng et al., 2018), leads to a large number of high-cost queries. In particular, it can be observed from Fig.1 that at least 50% of the queries made during these attacks are high-cost. To address this, they replaced the binary search with a search strategy inspired by the classic egg dropping problem (Alves et al., 2024), which is similar to a line search algorithm. However, they did not provide a *stealthy* variant of the Monte Carlo gradient estimation used in HSJA (Chen et al., 2020), GeoDA (Rahmati et al., 2020), and qFool (Liu et al., 2019b). Instead, they substituted it with an OPT-style gradient estimation procedure (Cheng et al., 2018).

Although stealthy attacks move toward addressing asymmetric query costs, they are not designed to handle arbitrary cost ratios. They implicitly assume that benign queries have zero cost, which may not reflect realistic settings where even benign queries contribute to the overall cost. In addition, since stealthy attacks could not adapt the Monte Carlo gradient approximation used in HSJA (Chen et al., 2020), they instead rely on a suboptimal and inefficient OPT-style gradient estimation (Cheng et al., 2018), which is already significantly outperformed by the HSJA gradient approximation under **symmetric** cost settings. These limitations motivate us to answer the following question:

> **Q**: How can we develop an **efficient** framework to adapt attacks for any **arbitrary** cost ratio **without discarding** any of their core components, including gradient estimation and binary search?

In this work, we propose a general framework for decision-based attacks that operates under arbitrary query cost asymmetries. Instead of only considering high-cost queries, we change the core components of black-box attacks, namely search along adversarial paths and gradient estimation, to explicitly reduce the *total query cost*. Our framework handles any cost ratio between high-cost and low-cost queries and completely outperforms stealthy attacks by optimizing the attack structure without sacrificing efficiency. Unlike stealthy attacks (Debenedetti et al., 2024), which rely on inefficient gradient approximations, we retain the more efficient Monte Carlo gradient estimation technique used in HSJA (Chen et al., 2020), GeoDA (Rahmati et al., 2020), and qFool (Liu et al., 2019b).

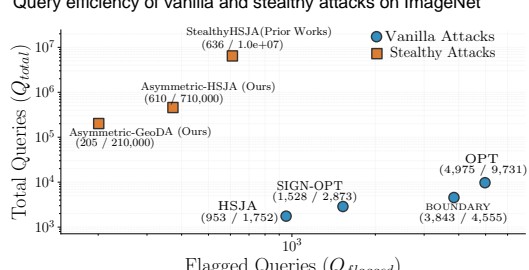

Figure 1: Each point represents the median number of queries required by an attack method to reach a median $\ell_2$ norm of 10. The x-axis shows the number of flagged queries ($Q_{\text{flagged}}$) and the y-axis reports the total number of queries ($Q_{\text{total}}$). It demonstrates the superiority of our method in achieving a more favorable trade-off between flagged and total number of queries in stealthy attack settings.

First, for the boundary search, instead of dividing the interval into two equal parts at each iteration, as in the standard binary search, we take a more conservative strategy. Specifically, we split the interval according to the cost ratio between high-cost and low-cost queries. This approach minimizes the expected cost rather than merely minimizing the expected number of queries. We call this method **A**symmetric **S**earch (AS).

Second, unlike traditional gradient estimation where samples are drawn from a norm ball centered at a boundary point, which causes roughly half the queries to be high-cost and the other half low-cost as in standard HSJA, we shift the center to a point in the low-cost region and generate queries around it (Fig. 2). This adjustment naturally reduces the frequency of high-cost queries, with the degree of shifting providing direct control over this frequency. To further reduce variance in estimation, we weight high-cost and low-cost queries differently when computing the gradient. We refer to this method as **A**symmetric **GR**adient **EST**imation (AGREST). Our framework is broadly compatible with a wide range of state-of-the-art decision-based attacks, including HSJA (Chen et al., 2020), GeoDA (Rahmati et al., 2020), and CGBA (Reza et al., 2023). Through both theoretical analysis and extensive experimental evaluation, we show that AGREST consistently outperforms existing attacks

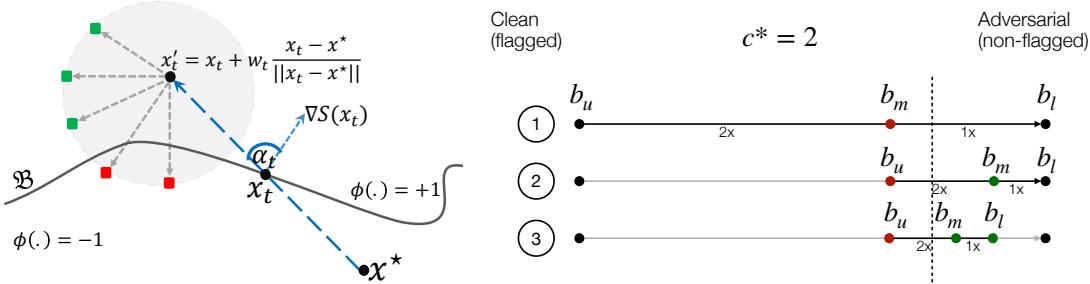

Figure 2: **Left.** Illustration of Asymmetric Gradient Estimation (AGREST), which reduces the frequency of high-cost queries by shifting the sampling region from $\mathbf{x}_t$ toward the adversarial region $\mathbf{x}'_t$ and appropriately reweighting the outcomes. **Right.** Three steps of Asymmetric Search (AS) along the path from a clean (flagged) source image to an adversarial (non-flagged) image. Flagged queries are shown in red, non-flagged queries in green, and the dashed line denotes the decision boundary.

under arbitrary cost ratios. Notably, even under extreme asymmetry where the cost of high-cost queries approaches infinity, our method achieves significantly lower total query costs to reach a given adversarial perturbation size compared to prior stealthy methods (Debenedetti et al., 2024) (see Fig. 1 (left)). This robustness underscores the effectiveness of our framework in balancing query efficiency and perturbation quality across diverse attack scenarios.

The contribution of our paper is as follows:

- To the best of our knowledge, we are the first to propose a versatile framework capable of handling arbitrary query cost ratios, providing flexibility across a wide range of scenarios.
- Our framework introduces **AS** and **AGREST** as two core operations that enhance existing algorithms. We conduct a comprehensive theoretical analysis to establish the foundations of the framework and demonstrate its robustness across diverse setups.
- We validate the framework through extensive empirical testing on benchmark datasets and models, including ImageNet, as well as advanced models such as CLIP and Vision Transformers and ResNet. This validation highlights the framework's superior performance.

## 2 PROBLEM STATEMENT

**An insight into unequal queries.** Consider an attacker trying to deceive an NSFW detector using decision-based methods. It may seem sufficient to choose an existing attack algorithm and add a small perturbation to an NSFW image based on that algorithm. However, this approach may encounter some practical obstacles. Most social networks enforce policies against uploading adult content, suspending users for violating these terms multiple times (Twitter, 2024a). Using the terminology from Debenedetti et al. (2024), this means that the cost of queries identified by the detector as NSFW, i.e., **flagged queries**, is higher than that of other queries, i.e., **non-flagged queries**. For example, on $\mathbb{X}$, an attacker can make up to 2,400 posts per day on a single account (Twitter, 2024b). However, after about 5 to 10 rule violations for uploading flagged posts, the attacker's account will be suspended, requiring them to create a new one. On the other hand, in existing decision-based attacks, approximately half of the made queries are flagged (Debenedetti et al., 2024). Therefore, if we assume the violation limit is 10, an attacker will be banned on $\mathbb{X}$ after about 20 posts. This example demonstrates the potential asymmetry in the costs of queries in a decision-based black-box setup. Debenedetti et al. (2024) addressed this asymmetry in costs by proposing stealthy attacks[1] designed to reduce the number of flagged queries. However, they overlooked the cost of non-flagged queries in their framework, leading to the generation of millions of non-flagged queries for every hundred flagged queries in stealthy attacks, which can also be costly.

For example, in the NSFW detector scenario, assume the attacker must create a new account after reaching the daily post limit. In stealthy attacks like HSJA, the attack can generate around $10^6$

---

[1]Hereafter, we refer to prior stealthy attacks simply as *stealthy attacks*, and to our approaches as *asymmetric attacks* to emphasize their cost-aware design. Though inherently stealthy due to *query cost awareness*, we adopt the term *asymmetric* attacks to distinguish our method from prior work (Debenedetti et al., 2024).

non-flagged queries for every 100 flagged queries (Fig. 1). While 100 flagged queries may lead to the creation of 10 new accounts, those $10^6$ non-flagged queries result in approximately 400 new accounts. This shows that non-flagged queries, despite being lower-cost, have a greater overall impact. Therefore, it is essential to develop generalized decision-based attacks that can effectively manage asymmetric query costs, making full use of low-cost queries without relying heavily on expensive ones.

**General formulation.** Assume that $f : \mathbb{R}^d \to \mathbb{R}^L$ is a pre-trained classifier with $L$ classes and parameters $\theta$. For an input image $\mathbf{x} \in [0, 1]^d$, $f_k(\mathbf{x})$, the $k^{th}$ component of $f(\mathbf{x})$, represents the predicted probability of the $k^{th}$ class. Additionally, for each correctly classified image $\mathbf{x}$ and query image $\mathbf{x}'$, we define

$$S_{\mathbf{x}}(\mathbf{x}') = \arg\max_{k \neq \hat{y}(\mathbf{x})} f_k(\mathbf{x}') - f_{\hat{y}(\mathbf{x})}(\mathbf{x}') \quad \text{and} \quad \phi_{\mathbf{x}}(\mathbf{x}') = \text{sign}\left(S_{\mathbf{x}}(\mathbf{x}')\right). \tag{1}$$

Given a correctly classified source image $\mathbf{x}^\star$, the attacker's goal is to find the closest perturbed image $\mathbf{x}'$ to the source image $\mathbf{x}^\star$ such that $\phi_{\mathbf{x}^\star}(\mathbf{x}') = 1$:

$$\underset{\mathbf{x}'}{\text{minimize}} \; \|\mathbf{x}^\star - \mathbf{x}'\| \quad \text{s.t.} \quad \phi_{\mathbf{x}^\star}(\mathbf{x}') = 1. \tag{2}$$

Note that in a decision-based black-box setup, the attacker has access to $\phi_{\mathbf{x}^\star}(\mathbf{x}')$ but not $S_{\mathbf{x}^\star}(\mathbf{x}')$. Previous methods sought to solve Eq. (2) while keeping the total number of queries as low as possible. However, as discussed, asymmetric query costs can make this approach ineffective. Instead, we must keep the total cost of queries in an asymmetric setup as low as possible (Debenedetti et al., 2024):

$$\text{cost} := Q_{\text{total}} \cdot c_0 \; + \; Q_{\text{flagged}} \cdot c_{\text{flagged}}, \tag{3}$$

where $Q_{\text{flagged}}$ is the number of flagged queries ($\phi(\mathbf{x}) = -1$), and $Q_{\text{total}}$ is the total number of queries. Our goal is to solve this for arbitrary values of $c_0$ and $c_{\text{flagged}}$, unlike stealthy attacks where $c_0 = 0$. Since only the relative magnitude of $c_{\text{flagged}}$ with respect to $c_0$ matters, we use the following reformulation to reduce the number of tunable parameters:

$$\text{cost} := Q_{\text{non-flagged}} \; + \; Q_{\text{flagged}} \cdot c^\star, \tag{4}$$

where $c^\star = \dfrac{c_{\text{flagged}} + c_0}{c_0}$. Existing decision-based attacks assume $c^\star = 1$, while stealthy attacks assume $c^\star = \infty$. Our goal in this paper is to propose a framework that is effective for any arbitrary value of $c^\star$, unlike both vanilla and stealthy attacks. Furthermore, we demonstrate that our approach outperforms stealthy attacks even when $c^\star = \infty$.

For brevity, we omit $\mathbf{x}^\star$ when mentioning $S$ and $\phi$. We also refer to queries $\mathbf{x}'$ where $\phi(\mathbf{x}') = -1$ as **high-cost queries** and others as **low-cost queries**. These new terms reflect the concept of general asymmetric costs better than the previous terms used by Debenedetti et al. (2024), i.e., flagged and non-flagged queries, which are more suitable when the discrepancy between query costs is too large.

## 3 PROPOSED METHOD

Decision-based black-box attacks typically involve two core operations, often applied iteratively to find small adversarial perturbations: 1. choosing a path, either straight, like GeoDA and HSJA (Rahmati et al., 2020; Chen et al., 2020), or curved, like SurFree and CGBA (Maho et al., 2021; Reza et al., 2023), and then 2. Searching along this path to find a new adversarial example, $\mathbf{x}_{t+1}$, that is closer to $\mathbf{x}^\star$ than $\mathbf{x}_t$, the adversarial example from the previous iteration. These attacks either choose a path randomly, as in Boundary Attack (Brendel et al., 2017) and SurFree, or use queries to find a path that leads to a closer adversarial example than a random path, as in HSJA, GeoDA, and CGBA. To find this better-than-random path, these attacks estimate the normalized gradient direction of $S$ at $\mathbf{x}_t$ by approximating $\nabla S(\mathbf{x}_t)$ as follows:

$$\widetilde{\nabla S}(\mathbf{x}_t) = \frac{1}{n_t} \sum_{i=1}^{n_t} \phi(\mathbf{x}_t + \delta \mathbf{u}_i) \mathbf{u}_i, \tag{5}$$

where $\delta$ is a small positive parameter and $\mathbf{u}_1, \ldots, \mathbf{u}_{n_t}$ are i.i.d. draws from either the uniform distribution over $\mathbb{S}^{d-1}$, the $(d-1)$-dimensional unit sphere, or the multivariate normal distribution. After finding a path, most attacks use variations of binary search to find $\mathbf{x}_{t+1}$. This is generally achieved by finding a boundary point $\mathbf{x}_{t+1}$ along the selected path, where $S(\mathbf{x}_{t+1}) = 0$, using binary search.

As highlighted by Debenedetti et al. (2024), the issue with binary search and Eq. (5) is that, in an asymmetric setup, crafting adversarial examples using these operations becomes costly because approximately half of the generated queries are high-cost. This raises the question of whether we can alter the distribution of generated queries to reduce the number of high-cost queries while maintaining the effectiveness of these two operations as observed in vanilla attacks. As a solution, we propose **AS** and **AGREST** techniques in the following sections.

## 3.1 ASYMMETRIC SEARCH (AS)

We define $T : [0, 1] \to \mathbb{R}^d$ as the function that parameterizes the search path. For example, when the path is a straight line between the source image $\mathbf{x}^\star$ and an adversarial image $\tilde{\mathbf{x}}$ Rahmati et al. (2020); Chen et al. (2020), the parameterization is given by $T(\theta) = \theta \mathbf{x}^\star + (1 - \theta) \tilde{\mathbf{x}}$. Similarly, when the search path follows a circular arc Reza et al. (2023); Maho et al. (2021) between $\mathbf{x}^\star$ and $\tilde{\mathbf{x}}$, lying on a circle in the 2D subspace spanned by $\mathbf{u} = \frac{\tilde{\mathbf{x}} - \mathbf{x}^\star}{\|\tilde{\mathbf{x}} - \mathbf{x}^\star\|_2}$ and a unit vector $\mathbf{v}$ satisfying $\langle \mathbf{u}, \mathbf{v} \rangle = 0$, the parameterization is given by $T(\theta) = \mathbf{x}^\star + \cos\left(\frac{\pi}{2}\theta\right) \cdot \|\tilde{\mathbf{x}} - \mathbf{x}^\star\|_2 \left(\cos\left(\frac{\pi}{2}\theta\right) \mathbf{u} + \sin\left(\frac{\pi}{2}\theta\right) \mathbf{v}\right)$.

As mentioned, the objective is to find the along-the-path boundary point $T(\theta^\star)$ for some $\theta^\star \in [0, 1]$. However, since the search space is discrete in practice, our goal is to find a point $\mathbf{x}_{t+1}$ that is near the boundary, i.e., where $S(\mathbf{x}_{t+1}) \approx 0$. This implies that for a given error threshold $\tau$, we need to find $[a, b] \subset [0, 1]$ such that $0 < b - a \leq \tau$ and $\theta^\star \in [a, b]$. Since $S$ is continuous, one approach to achieve this is to find $0 \leq k \leq \lceil \frac{1}{\tau} \rceil$ for which $\phi(T(k\tau)) = 1$ and $\phi(T((k+1)\tau)) = -1$. The smaller $\tau$ is, the closer we get to the boundary, although this requires more queries.

It is well-known that in one-dimensional search, binary search is the optimal comparison-based algorithm in terms of minimizing the expected number of queries, assuming that the boundary point is uniformly distributed along the path (see

---

**Algorithm 1** AS Algorithm

**Inputs:** Parametrization function $T$, threshold $\tau$

**Outputs:** Near-boundary adversarial example

1: $b_l \leftarrow 0$, $b_u \leftarrow \lceil * \rceil \frac{1}{\tau}$
2: **while** $b_u - b_l > 1$ **do**
3:     $b_m \leftarrow b_l + \lceil * \rceil \frac{b_u - b_l}{(c^\star + 1)}$
4:     **if** $\phi(T(b_m\tau)) = 1$ **then**
5:         $b_l \leftarrow b_m$
6:     **else**
7:         $b_u \leftarrow b_m$
8:     **end if**
9: **end while**
10: **return** $T(b_l\tau)$

---

Assumption A1). Nonetheless, in the asymmetric cost setting, the expected cost of binary search is $\Theta(c^\star \log(1/\tau))$, because the expected number of queries is $\Theta(\log(1/\tau))$, and about half of these queries are expected to incur the higher cost $c^\star$.

**Assumption A1.** *Let $\Theta^\star_{rv} \in [0, 1)$ be a random variable. If $T(\Theta^\star_{rv})$ lies on the decision boundary, that is, $S(T(\Theta^\star_{rv})) = 0$, then we assume $\Theta^\star_{rv}$ is drawn uniformly from $[0, 1)$.*

The core idea behind AS is similar to that of binary search, but with a more conservative strategy to account for asymmetric costs. Instead of splitting the interval into two equal parts, AS divides it with a $1 : c^\star$ ratio at each step, favoring lower-cost queries. More specifically, suppose we know the desired point lies within $[b_l\tau, b_u\tau] \subset [0, 1]$. Then, as shown in Alg. 1, if $\phi\left(T\left(b_l\tau + \left\lceil \frac{b_u - b_l}{c^\star + 1} \right\rceil \tau\right)\right) = 1$, AS continues the search in $\left[b_l\tau + \left\lceil \frac{b_u - b_l}{c^\star + 1} \right\rceil \tau, b_u\tau\right]$; otherwise, it proceeds within $\left[b_l\tau, b_l\tau + \left\lceil \frac{b_u - b_l}{c^\star + 1} \right\rceil \tau\right]$. This process is repeated until AS locates a point within $\tau$ of the boundary.

Note that when $c^\star = 1$, AS reduces to standard binary search, and when $c^\star = \infty$, it becomes a simple line search strategy, as used in stealthy attacks (Debenedetti et al., 2024), where the algorithm checks $\tau, 2\tau, 3\tau, \ldots$ sequentially. The expected cost of AS is given in Thm. 1, showing that it improves over binary search by a factor of $\Theta(\log(c^\star + 1))$.

**Theorem 1.** *(Cost Analysis of AS) Suppose $0 < \tau < 1$ and $c^\star \geq 1$. Under Assumption A1, the expected cost of the AS algorithm is $\mathcal{O}(c^\star \log_{(c^\star+1)}(1/\tau))$.*

To illustrate the effect of AS in practice, we compare the cost of AS and binary search when $c^\star = 10^3$, and we observe that the cost of binary search is approximately 2.5 times higher than that of AS. The results are provided in App. C. An illustration of the AS algorithm can be found in App. I (Right) and Fig. 9.

## 3.2 ASYMMETRIC GRADIENT ESTIMATION (AGREST)

As mentioned earlier, our goal is to adjust the distribution of queries generated during the process. To achieve this, we first propose a family of estimations that provides flexibility in adjusting the distribution of queries used for estimation. Then, we introduce a method to select the estimator within this family that maximizes the similarity between the approximated gradient and the true gradient.

**How can we control the query distribution?** The main idea behind AGREST is to estimate $\nabla S(\mathbf{x}_t)$ by approximating the gradient at a point like $\mathbf{x}_t'$, which is close to $\mathbf{x}_t$. This approach allows us to alter the distribution of made queries while approximating effective directions for the attacks. In this method, we move $\mathbf{x}_t$ away from $\mathbf{x}^\star$ by $\omega_t$, the overshooting value, to reach the new point $\mathbf{x}_t' = \mathbf{x}_t + \omega_t \frac{\mathbf{x}_t - \mathbf{x}^\star}{\|\mathbf{x}_t - \mathbf{x}^\star\|_2}$ (Fig. 2 (left)). Then, we use Eq. (5) at $\mathbf{x}_t'$ almost similar to the vanilla estimation, except that AGREST assigns more weight to high-cost queries than to low-cost ones (using importance sampling). In other words, we estimate $\nabla S$ as follows:

$$\widehat{\nabla S}(\mathbf{x}_t, \omega_t, \beta_t) = \frac{1}{n_t} \sum_{i=1}^{n_t} \widehat{\phi}_t(\mathbf{x}_t' + \delta \mathbf{u}_i) \mathbf{u}_i = \frac{1}{n_t} \sum_{i=1}^{n_t} \widehat{\phi}_t \left( \mathbf{x}_t + \omega_t \frac{\mathbf{x}_t - \mathbf{x}^\star}{\|\mathbf{x}_t - \mathbf{x}^\star\|_2} + \delta \mathbf{u}_i \right) \mathbf{u}_i, \quad (6)$$

where $\widehat{\phi}_t(\mathbf{x}) = (1 - \beta_t)\mathbf{1}\{\phi(\mathbf{x}) = 1\} - \beta_t \mathbf{1}\{\phi(\mathbf{x}) = -1\}$ is the sampling weight function, $\frac{1}{2} \leq \beta_t < 1$ is the sampling weight parameter, $\mathbf{u}_1, \ldots, \mathbf{u}_{n_t}$ are i.i.d. draws from UNIFORM($\mathbb{S}^{d-1}$), and $\mathbf{1}\{\cdot\}$ denotes the indicator function. The parameters $\omega_t$ and $\beta_t$ allow us to control the likelihood of making high-cost queries and their associated weight in our estimation.

**How can we choose the best AGREST estimator?** To ensure the selected estimation is as close as possible to the true gradient direction among all AGREST estimators, one potential solution is to find the estimator that maximizes:

$$\mu(\mathbf{x}_t, \omega_t, \beta_t, n_t) = \mathbb{E}_{\mathbf{u}_{1:n_t}} \cos\left(\nabla S(\mathbf{x}_t), \widehat{\nabla S}(\mathbf{x}_t, \omega_t, \beta_t)\right). \quad (7)$$

within the query budget, where $\cos$ represents the cosine similarity function. To calculate this function, we need to assume that $S$ has certain characteristics. One common choice is to assume that $S$ is $L$-smooth. However, this assumption introduces excessive complexity to our analysis and may add additional hyperparameters related to $L$ to the current set of hyperparameters in the existing attacks. Therefore, similar to Rahmati et al. (2020); Maho et al. (2021) and based on observations from Fawzi et al. (2016), we assume that $S$ is locally linear around $\mathbf{x}_t$, $S(\mathbf{x}_t' + \delta \mathbf{u}) \approx S(\mathbf{x}_t) + \langle \nabla S(\mathbf{x}_t), \mathbf{x}_t' + \delta \mathbf{u} - \mathbf{x}_t \rangle$. Since $\mathbf{x}_t$ is a boundary point, $S(\mathbf{x}_t) = 0$. Thus, we have:

$$\phi(\mathbf{x}_t' + \delta \mathbf{u}) \approx \text{sign}\left(\langle \nabla S(\mathbf{x}_t), \mathbf{x}_t' + \delta \mathbf{u} - \mathbf{x}_t \rangle\right) = \text{sign}(\cos \alpha_t \cdot \omega_t + \langle \mathbf{g}_t, \delta \mathbf{u} \rangle), \quad (8)$$

where $\mathbf{g}_t = \frac{\nabla S(\mathbf{x}_t)}{\|\nabla S(\mathbf{x}_t)\|_2}$ and $\alpha_t$ is the angle between $\mathbf{x}_t - \mathbf{x}^\star$ and $\mathbf{g}_t$ (Fig. 2). Additionally, based on this assumption, we can calculate the probability of low-cost queries, namely $p_t(\omega_t) = \mathbb{P}[\phi(\mathbf{x}_t' + \delta \mathbf{u}) = 1]$, using Lem. 1.

**Lemma 1.** *(Hyperspherical Cap Chudnov (1986))* *Under local linearity around* $\mathbf{x}_t$ *(Eq. (8)), we have* $p_t(\omega_t) = \frac{1}{2}\left(1 + \mathcal{I}_{d-2}(\delta^{-1} \cos \alpha_t \omega_t)/\mathcal{I}_{d-2}(0)\right)$*, where* $\mathcal{I}_d(x) = \int_0^{1-x}(1-t^2)^{(d-1)/2}dt$.

Based on Lem. 1, we can infer that $p_t(\omega_t)$ is strictly increasing and therefore invertible. Nonetheless, even with the linearity assumption, calculating the expected value remains challenging due to the nonlinearity of cosine similarity and the complexity of handling multiple independent random vectors. Therefore, inspired by measure concentration Ledoux (2001), we approximate $\mu(\mathbf{x}_t, \omega_t, \beta_t, n_t)$ as follows:

$$J(\mathbf{x}_t, \omega_t, \beta_t, n_t) = \left(n_t^{1/2} \cdot \mathbb{E}\left[\widehat{\phi}(\mathbf{x}_t' + \delta \mathbf{u})\langle \mathbf{g}_t, \mathbf{u} \rangle\right]\right) \cdot \left(\mathbb{E}\left[\widehat{\phi}(\mathbf{x}_t' + \delta \mathbf{u})^2\right]\right)^{-1/2}. \quad (9)$$

This new objective is easier to calculate since it removes the need to deal with multiple random vectors. Thm. 2 establishes a convergence bound for the approximation.

**Theorem 2.** *(Expected Cosine Similarity Approximation)* *Under the local linearity assumption around* $\mathbf{x}_t$*, for any constants* $0 < z < \frac{1}{8}$ *and* $\frac{1}{2} \leq q, \beta < 1$*, as* $n_t$ *and* $d$ *approach infinity, we have*

$$\left|\frac{\mu(\mathbf{x}_t, p_t^{-1}(q), \beta, n_t)}{J(\mathbf{x}_t, p_t^{-1}(q), \beta, n_t)} - 1\right| \leq \mathcal{O}\left(d^{-z}\right) \quad (10)$$

.

Now, we can formulate the optimization problem. The goal is to maximize $J(\mathbf{x}_t, \omega_t, \beta_t, n_t)$ within a query budget. Specifically, we want to:

$$\max_{\omega_t, \beta_t, n_t} \quad J(\mathbf{x}_t, \omega_t, \beta_t, n_t) \qquad \text{s.t.} \quad n_t(c^\star - (c^\star - 1)p_t(\omega_t)) \leq c_t \tag{11}$$

where $c_t$ is the maximum allowed cost of estimation at iteration $t$ of the algorithm. The constraint in Eq. (11) ensures that the expected estimation cost at iteration $t$ is at most $c_t$. To solve this optimization problem, we propose Thm. 3.

**Theorem 3.** *(Optimal AGREST Parameters)* *Suppose the solution to Eq.* (11) *is represented by* $(\omega_t^\star, w_t^\star, n_t^\star)$. *Given the local linearity around* $\mathbf{x}_t$, *the following statements hold:*

1. $n_t^\star = c_t \left(c^\star - (c^\star - 1)p_t(\omega_t^\star)\right)^{-1}$ *and* $\beta_t^\star = p_t(\omega_t^\star)$.

2. $\omega_t^\star$ *maximizes the following function over the interval* $[0, \delta/\cos\alpha_t]$:

$$\widehat{J}_t(\omega_t) = \left(1 - \left(\delta^{-1}\cos\alpha_t\omega_t\right)^2\right)^{d-1} \left(p_t(\omega_t)\left(1 - p_t(\omega_t)\right)\left(c^\star - (c^\star - 1)p_t(\omega_t)\right)\right)^{-1} \tag{12}$$

An immediate consequence of Lem. 1 and Thm. 3 is the existence of $\omega^\star$ such that $\omega^\star = \cos\alpha_1 \cdot \omega_1^\star = \cos\alpha_2 \cdot \omega_2^\star = \ldots$. As a result, we aim to find $\omega^\star$. One problem is that $\mathcal{I}_d$ has a complex closed form. Thus, finding a closed form for $\omega^\star$ would be challenging. Instead, we use numerical methods to evaluate this integral and numerical optimization techniques to find $\omega^\star$. In particular, we use QUADPACK Piessens et al. (2012) for the integral calculation and the Nelder–Mead method Nelder & Mead (1965) for maximizing $\widehat{J}_t(\omega_t)$. Another problem is that we need to know $\alpha_t$, which is not possible in a black-box setup. Thus, we need to estimate $\alpha_t$ at each iteration $t$.

**How can we estimate** $\alpha_t$? We split this problem into two steps: 1. estimating $\alpha_1$, and 2. understanding the behavior of $\alpha_t$ with respect to $t$. For the first problem, initially, we expect $\mathbf{x}_1 - \mathbf{x}^\star$ and $\mathbf{g}_1$ to be somehow independent, as most existing attacks select $\mathbf{x}_1$ using a random direction. The only reasonable assumption about these two vectors is that they likely have a positive correlation, i.e., $\langle \mathbf{x}_1 - \mathbf{x}^\star, \mathbf{g}_1 \rangle \geq 0$. Specifically, if we know that $\mathbf{x}_1$ is the closest boundary point to $\mathbf{x}^\star$ along the

---

**Algorithm 2** Overshooting Scheduler Step

**Inputs:** Iteration $t$, dimension $d$, desired probability $p$, scheduler rate $m$
**Outputs:** Next cosine value $\alpha_{t+1}$
1: $\alpha_1 \leftarrow \text{INIT-ANGLE}\,(d)$      ▷ Thm. 4
2: $\hat{\alpha}_{t+1} \leftarrow 1 - (1 - \cos\alpha_1)(t+1)^{-m}$
3: $\alpha_{t+1} \leftarrow \arccos(\hat{\alpha}_{t+1})$
4: **return** $\alpha_{t+1}$

---

direction of $\mathbf{x}_1 - \mathbf{x}^\star$, meaning there is no $0 < r < 1$ such that $\phi(\mathbf{x}^\star + r(\mathbf{x}_1 - \mathbf{x}^\star)) = 1$, this assumption provably holds. Given this assumption, we can use Thm. 4 to attain $\alpha_1$.

**Theorem 4.** *(Initial Cosine Value)* *Under local linearity around* $\mathbf{x}_1$, *if there is no* $0 < r < 1$ *such that* $\phi(\mathbf{x}^\star + r(\mathbf{x}_1 - \mathbf{x}^\star)) = 1$, *then we have* $\mathbb{E}[\cos\alpha_1] = \Gamma\left(\frac{d}{2}\right)\left(2\sqrt{\pi}\,\Gamma\left(\frac{d+1}{2}\right)\right)^{-1}$.

The next step is to estimate $\alpha_t$ after the first iteration. Chen et al. Chen et al. (2020) showed that in HSJA, $\cos(\mathbf{x}_t - \mathbf{x}^\star, \mathbf{g}_t) \geq 1 - c\,t^{-m}$ for some constant $c$ and $0 < m < \frac{1}{2}$. This motivated us to heuristically estimate $\alpha_t$ as $\arccos(1 - (1 - \cos\alpha_1)t^{-m})$, where $m$ is a new hyperparameter introduced to the existing attacks (Alg. 2). As stated in Thm. 3, under the assumption of local linearity, the value of $\beta_t$ in an optimal AGREST estimation is the probability of making low-cost queries. However, similar to Chen et al. (2020), we use the empirical probability of making low-cost queries, namely $\frac{n_L}{n_L + n_H}$, to reduce the variance of the estimation. Here, $n_H$ and $n_L$ represent the number of high-cost and low-cost queries made in an AGREST estimator, respectively. Additionally, we set $c_t$ to the expected cost of the vanilla attack, namely $\frac{n_t'(c^\star + 1)}{2}$, where $n_t'$ is the number of made queries by the vanilla attack at iteration $t$. A detailed overview of AGREST is provided in Alg. 3. Note that in practice, most attacks are performed in a given subspace rather than in the entire space to improve sample efficiency. In these cases, we use the effective dimension $d'$ of the subspace instead of $d$, the dimension of the original space. For more details, see App. D.3. Finally, it is worth mentioning that the probability of making low-cost queries in AGREST closely follows our theoretical analysis in practice. Further details and empirical results are presented in App. C.

---

**Algorithm 3** AGREST Estimation

---

**Inputs:** Iteration $t$, source image $\mathbf{x}^\star$, boundary point $\mathbf{x}_t$, dimension $d$, high-cost query cost $c^\star$, sampling radius $\delta$, sampling batch size $b$, cosine value $\alpha_t$, vanilla gradient estimation query budget $n'_t$, scheduler rate $m$

**Outputs:** Normalized approximated direction $g_t$, next cosine value $\alpha_{t+1}$

1: $n_L \leftarrow 0, \; n_H \leftarrow 0, \; \mathbf{v}^+ \leftarrow \vec{0}, \; \mathbf{v}^- \leftarrow \vec{0}, \; \hat{c} \leftarrow 0, \; \omega^\star \leftarrow \text{OVERSHOOTING}\,(c^\star)$      ▷ Thm. 3
2: $\omega_t \leftarrow \omega^\star / \cos \alpha_t, \; c_t \leftarrow n'_t(c^\star + 1)/2$
3: **while** $\hat{c} < c_t$ **do**
4:      **for each** $\mathbf{u}_i \sim \text{UNIFORM}(\mathbb{S}^{d-1}), \, i = 1, \ldots, b$ **do**
5:          **if** $\phi \left( \mathbf{x}_t + \omega_t \frac{\mathbf{x}_t - \mathbf{x}^\star}{\|\mathbf{x}_t - \mathbf{x}^\star\|_2} + \delta \mathbf{u}_i \right) = 1$ **then**
6:              $\mathbf{v}^+ \leftarrow \mathbf{v}^+ + \mathbf{u}_i, \; n_L \leftarrow n_L + 1, \; \hat{c} \leftarrow \hat{c} + 1$
7:          **else**
8:              $\mathbf{v}^- \leftarrow \mathbf{v}^- - \mathbf{u}_i, \; n_H \leftarrow n_H + 1, \; \hat{c} \leftarrow \hat{c} + c^\star$
9:          **end if**
10:      **end for**
11: **end while**
12: $\hat{p}_t \leftarrow n_L / (n_L + n_H)$
13: $g_t \leftarrow (1 - \hat{p}_t)\mathbf{v}^+ + \hat{p}_t\mathbf{v}^-, \; \alpha_{t+1} \leftarrow \text{SCHEDULER-STEP}\,(t, \hat{p}_t, m)$      ▷ Alg. 2
14: **return** $g_t / \|g_t\|_2, \; \alpha_{t+1}$

---

## 4 EXPERIMENTS

**Model, dataset, and metric.** We employed ImageNet-trained models: ResNet-50 (He et al., 2016), ViT-B/32, ViT-B/16 (Dosovitskiy et al., 2021), and CLIP (Radford et al., 2021). Original images ($\mathbf{x}^\star$) were 500 correctly classified ImageNet validation samples similar to the Debenedetti et al. (2024). Numerical tasks used SciPy (Virtanen et al., 2020). Attack performance was measured by the median $\ell_2$ distance between perturbations and originals over query costs, consistent with previous work.

**Attacks and hyperparameters.** We modify SurFree, HSJA, GeoDA, and CGBA by using AS for search and AGREST for gradient estimation where applicable. The other components of the attacks remain largely unchanged (see App. D). Moreover, to compare our framework with stealthy attacks, we use Stealthy HSJA, as it outperforms other stealthy attacks when the $\ell_2$ norm is the evaluation metric (Fig. 3 of Debenedetti et al. (2024)). For the hyperparameters used in the vanilla attacks, we generally use the same values. The only exception is the subspace method in SurFree. Specifically, instead of using the $\text{DCT}_{8\times8}$ method in SurFree, we set it to $\text{DCT}_{\text{full}}$. This adjustment allows for a fair comparison of SurFree with other attacks, as GeoDA and CGBA both use the $\text{DCT}_{\text{full}}$ technique. Furthermore, we set the newly introduced hyperparameter $m$, the overshooting scheduler rate, to 0.02, 0.06, and 0.06 for HSJA, GeoDA, and CGBA, respectively (for more details see App. D.2).

**Ablation study.** To evaluate the effectiveness of AS and AGREST in different attacks, we test various combinations of these two approaches with each attack when $c^\star = 2, 5, 10^2$, or $10^3$ (Tab. 1). As shown in Tab. 1, for SurFree, we compare the vanilla attack with **A-SurFree**, the new asymmetric attack that utilizes AS. As expected, replacing binary search with AS leads to smaller adversarial perturbations for all $c^\star$. Furthermore, we compare the performance of gradient-based attacks with their corresponding variations, namely: 1. Replacing binary search with AS 2. Replacing vanilla gradient estimation with AGREST 3. Combining the two previous approaches to obtain **A-HSJA**, **A-GeoDA**, **A-CGBA**. In general, using asymmetric attacks results in incremental improvements when $c^\star = 2$ or 5. This is expected because we anticipate binary search and vanilla gradient estimation perform well when the cost of high-cost queries is not significantly different from the cost of low-cost queries. However, for larger values of $c^\star$, namely $10^2$ and $10^3$, the improvements are substantial. Specifically, using AGREST alone reduces the $\ell_2$ norm by approximately 40% in all cases. Moreover, combining AGREST with AS further decreases the norm. One notable point is that AGREST enhances attacks utilizing gradient estimation more than AS does. This occurs because attacks using gradient estimation spend most of their query budget on gradient approximation rather than on search. (Tab. III of Debenedetti et al. (2024)). We also compared several methods, including some baseline transfer attacks, using attack success rate (ASR) as the metric (see App. E).

**Comparison to *stealthy* attacks.** As mentioned, for larger values of $c^\star$, we expect asymmetric attacks to significantly improve over vanilla attacks. Nonetheless, in these cases, we must compare

Table 1: Median $\ell_2$ distance for various $c^\star$ values and different types of attacks across neural network architectures. VA stands for Vanilla Attack. The bold numbers represent the best performance among different variants of each attack for each $c^\star$ value and model (For a comprehensive analysis of attacks under varying total cost constraints, we refer readers to Tab. 9 and Tab. 10 in App. F, which present exhaustive experimental results across different total cost budgets and query cost $c^\star$.)

| | | ResNet-50 | | | | ViT-B/32 | | | |
|---|---|---|---|---|---|---|---|---|---|
| | | $c^\star = 2$ | $c^\star = 5$ | $c^\star = 10^2$ | $c^\star = 10^3$ | $c^\star = 2$ | $c^\star = 5$ | $c^\star = 10^2$ | $c^\star = 10^3$ |
| | | *Total Cost of Queries* | | | | | | | |
| *Attack* | *Method* | 10K | 15K | 150K | 250K | 10K | 15K | 150K | 250K |
| SurFree | VA | 4.09 | 5.19 | 5.21 | 17.49 | 2.9 | 2.5 | 5.13 | 16.12 |
| | VA+AS (A-SurFree) | **3.45** | **3.52** | **3.80** | **7.59** | **2.4** | **2.1** | **3.68** | **6.35** |
| HSJA | VA | 2.24 | 2.77 | 4.66 | 23.72 | 18.3 | 13.9 | 4.21 | 22.46 |
| | VA+AS | 2.16 | 2.72 | 4.09 | 19.07 | 18.3 | 13.5 | 3.88 | 18.79 |
| | VA+AGREST | 2.19 | 2.51 | 2.49 | 14.62 | 2.7 | 2.2 | 2.19 | 11.28 |
| | VA+AS+AGREST (A-HSJA) | **2.13** | **2.39** | **2.16** | **12.28** | **2.7** | **2.1** | **2.06** | **10.74** |
| GeoDA | VA | 2.80 | 3.21 | 4.02 | 10.80 | 2.7 | 2.4 | 3.97 | 9.83 |
| | VA+AS | **2.66** | 3.12 | 3.32 | 9.24 | 2.7 | 2.3 | 3.12 | 8.78 |
| | VA+AGREST | 2.89 | 2.95 | 2.19 | 6.28 | **1.9** | **1.7** | 2.10 | 5.12 |
| | VA+AS+AGREST (A-GeoDA) | 2.93 | **2.8** | **2.11** | **5.78** | 1.9 | 1.8 | **2.03** | **4.35** |
| CGBA | VA | 1.21 | 1.42 | 2.22 | 9.97 | 1.6 | 1.4 | 2.13 | 9.67 |
| | VA+AS | 1.17 | 1.39 | 2.06 | 9.28 | 1.7 | 1.3 | 1.97 | 8.24 |
| | VA+AGREST | **1.12** | 1.36 | 1.63 | **5.73** | 1.5 | 1.3 | 1.56 | 5.46 |
| | VA+AS+AGREST (A-CGBA) | 1.15 | **1.33** | **1.58** | 6.23 | **1.5** | **1.2** | **1.42** | **5.61** |

our framework to stealthy attacks, since, unlike with lower to medium values of $c^\star$, stealthy attacks outperform vanilla attacks when $c^\star$ is large (Fig. 7 of Debenedetti et al. (2024)). As a result, we evaluate the performance of A-SurFree, A-HSJA, A-GeoDA, and A-CGBA against Stealthy HSJA on the ResNet model when $c^\star = 10^4$, $10^5$, or $\infty$. As demonstrated in Fig. 3, when $c^\star = 10^4$, all asymmetric attacks, including A-HSJA which retains the gradient estimation method that Stealthy HSJA discards, outperform Stealthy HSJA. The same holds for $c^\star = 10^5$ (Fig. 3). For the case where $c^\star = \infty$, following Debenedetti et al. (2024), we determine the cost of each attack by counting the number of high-cost (flagged) queries it generates. In this setup, we assume $c^\star = 10^5$ during the execution of AGREST and AS. As shown in Fig. 3, all asymmetric attacks outperform Stealthy HSJA by a wide margin.

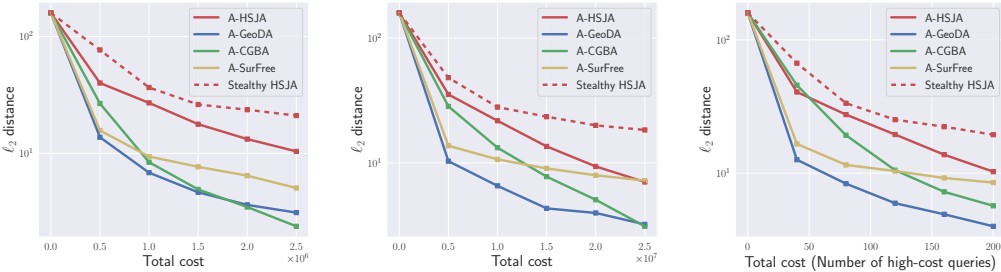

Figure 3: **Performance of various asymmetric attacks compared to Stealthy HSJA under high cost asymmetry with ResNet-50.** The value of $c^\star$ is $10^4$, $10^5$, and $\infty$ from left to right.

**Asymmetric attacks against CLIP.** We evaluate CLIP (Radford et al., 2021) as a representative vision-language model (VLM) under both zero-shot and fine-tuned settings. Our asymmetric attack achieves significantly better performance than stealthy baselines; results are provided in App. H.

## 5 CONCLUSION AND OUTLOOK

We proposed a framework that extends existing decision-based black-box attacks to handle asymmetric query costs, where querying the source class is more expensive than others. Our method introduces new gradient estimation and search techniques, achieving significant improvements over both standard and stealthy attack baselines. However, it introduces a new hyperparameter, which may require tuning for different settings. Additionally, we assume local linearity around decision boundaries; while this assumption is common in the adversarial examples literature, it may not

hold in practice. There are also many interesting directions for future work, such as generalizing the framework beyond the binary setting of source versus non-source classes. For instance, different target classes may each have their own associated query cost. Applying our framework to vision-language models such as Vision LLaMA (Chu et al., 2024) is another promising direction. AS could also enhance jailbreak attacks on large language models, potentially replacing random search-based methods (Andriushchenko et al., 2024; Chao et al., 2024), though adapting our framework to LLMs presents challenges due to the discrete nature of text prompts (Rocamora et al., 2025). Exploring these avenues could expand the impact of asymmetric attacks across a wide range of applications.

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

# A   RELATED WORK

**Decision-based attacks.**   Adversarial examples can be crafted in three setups: white-box (Goodfellow et al., 2014; Moosavi-Dezfooli et al., 2016; Carlini & Wagner, 2017; Abdolahpourrostam et al., 2024), score-based black-box (Narodytska & Kasiviswanathan, 2016; Chen et al., 2017; Ilyas et al., 2018), and decision-based black-box (Brendel et al., 2017). In the decision-based black-box setup, the attacker relies solely on predictions without access to models or class scores. Moreover, decision-based attacks can be either targeted or non-targeted. Non-targeted attacks craft adversarial examples without any constraints on the model's prediction for the adversarial example. Boundary Attack (Brendel et al., 2017) was the first effective decision-based attack based on random walking. OPT (Cheng et al., 2018) outperforms Boundary Attack by introducing a gradient-based approach. Inspired by zeroth-order optimization methods (Flaxman et al., 2004; Nesterov & Spokoiny, 2017), HSJA estimates the gradient of the classification margin without direct access to the margin itself and minimizes perturbation size using optimization techniques (Chen et al., 2020). QEBA (Li et al., 2020) uses various techniques to approximate the gradient of the classification margin more effectively than HSJA, leveraging insights like local similarity and the importance of the low-frequency subspace. GeoDA and qFool (Rahmati et al., 2020; Liu et al., 2019a) use techniques similar to HSJA's gradient estimation to locally approximate the decision boundary as a hyperplane at each iteration. They then search for optimal adversarial examples based on these estimated hyperplanes. While gradient-based methods outperform previous approaches, their reliance on generating numerous queries for efficient gradient estimation led (Maho et al., 2021) to focus on decision boundary geometry. They introduced SurFree, which iteratively selects a random 2D subspace and searches for adversarial examples along a circular path. In a similar way, TriA (Wang et al., 2022) generates effective adversarial examples while using minimal queries. CGBA (Reza et al., 2023) combines the gradient approximation method from GeoDA with SurFree's 2D subspace search technique to achieve state-of-the-art results.

**Asymmetric query costs.**   Existing decision-based attacks assume all queries have the same cost. However, (Debenedetti et al., 2024) showed this may not be the case in real-world scenarios. They found that queries belonging to the target class can be problematic in certain situations and noted that all decision-based attacks produce many of these bad queries. To mitigate this, they introduced stealthy attacks inspired by the egg dropping problem (Alves et al., 2024). However, stealthy attacks face two main challenges. First, they overlook the cost of queries that are not bad. For example, their most effective attack, Stealthy HSJA, generates about $10^7$ queries for every 1,000 bad queries. Second, to reduce the number of bad queries during the gradient estimation phase, (Debenedetti et al., 2024) replaced the HSJA gradient approximation, known for its benefits in crafting adversarial examples, with OPT gradient estimation, believing that modifying the HSJA gradient estimation to perform well in this new setup would be difficult. In this paper, we address these challenges in non-targeted decision-based attacks. In particular, we find a way to efficiently distribute our total query budget between problematic and non-problematic queries while keeping the HSJA method of gradient approximation by slightly modifying the method.

## B PROOFS

### B.1 PROOF OF THM. 1

We define the expected cost of the algorithm for $1 < m < \left\lceil \frac{1}{\tau} \right\rceil$ points as $C(m)$. Using Alg. 1 and applying the law of total expectation, we obtain

$$C(m) = \mathbb{P}\left[\phi\left(T\left(b_m\tau\right)\right) = -1\right]\left(C\left(\frac{m}{c^\star + 1}\right) + c^\star\right) + \mathbb{P}\left[\phi\left(T\left(b_m\tau\right)\right) = 1\right]\left(C\left(\frac{c^\star m}{c^\star + 1}\right) + 1\right). \tag{13}$$

We now claim that

$$C(m) < 2c^\star \left\lceil \log_{c^\star+1} m \right\rceil. \tag{14}$$

We prove this by induction.

For the base case, when $m \leq c^\star + 1$, Alg. 1 reduces to a simple line search. Assuming a uniform distribution for the boundary point, the expected cost of the line search is approximately $c^\star + m/2$, which clearly satisfies Eq. (14).

For the induction step, suppose the claim holds for all values smaller than $m$. Under the uniform distribution assumption for the boundary point, we have

$$\mathbb{P}\left[\phi\left(T\left(b_m\tau\right)\right) = -1\right] = \frac{1}{c^\star + 1}, \quad \mathbb{P}\left[\phi\left(T\left(b_m\tau\right)\right) = 1\right] = \frac{c^\star}{c^\star + 1}.$$

Substituting into Eq. (13) and applying the induction hypothesis yields:

$$C(m) = \frac{1}{c^\star + 1}\left(C\left(\frac{m}{c^\star + 1}\right) + c^\star\right) + \frac{c^\star}{c^\star + 1}\left(C\left(\frac{c^\star m}{c^\star + 1}\right) + 1\right)$$
$$< \frac{1}{c^\star + 1}\left(2c^\star \left\lceil \log_{c^\star+1} \frac{m}{c^\star + 1} \right\rceil + c^\star\right) + \frac{c^\star}{c^\star + 1}\left(2c^\star \left\lceil \log_{c^\star+1} \frac{c^\star m}{c^\star + 1} \right\rceil + 1\right).$$

Noting that

$$\log_{c^\star+1}\left(\frac{m}{c^\star + 1}\right) = \log_{c^\star+1}(m) - 1, \quad \log_{c^\star+1}\left(\frac{c^\star m}{c^\star + 1}\right) = \log_{c^\star+1}(m) + \log_{c^\star+1}\left(\frac{c^\star}{c^\star + 1}\right),$$

and observing that $\log_{c^\star+1}\left(\frac{c^\star}{c^\star+1}\right) < 0$ but close to 0 for large $c^\star$, we can bound both ceilings by $\left\lceil \log_{c^\star+1}(m) \right\rceil$.

Thus,

$$C(m) < \frac{2c^\star}{c^\star + 1}\left(\left\lceil \log_{c^\star+1} m \right\rceil - 1\right) + \frac{c^\star}{c^\star + 1}c^\star + \frac{2c^{\star 2}}{c^\star + 1}\left\lceil \log_{c^\star+1} m \right\rceil + \frac{c^\star}{c^\star + 1}$$
$$= 2c^\star \left\lceil \log_{c^\star+1} m \right\rceil.$$

Thus, the expected complexity of the algorithm is

$$\mathcal{O}\left(\frac{c^\star \log\left(1/\tau\right)}{\log(c^\star + 1)}\right),$$

completing the proof.

∎

### B.2 PROOF OF THM. 2

Before proving the theorem, we first introduce some useful lemmas.

**Lemma 2.** *(Lévy's Lemma (Milman & Schechtman, 1986; Ledoux, 2001)) Let $f : \mathbb{S}^{d-1} \to \mathbb{R}$ be an L-Lipschitz function on the unit hypersphere, and let $\mathbf{x} \sim \text{UNIFORM}\left(\mathbb{S}^{d-1}\right)$. Then, for some constant $C > 0$, we have:*

$$\mathbb{P}\left(|f\left(\mathbf{x}\right) - \mathbb{E}f\left(\mathbf{x}\right)| > \varepsilon\right) \leq 2\exp\left(-\frac{Cd\varepsilon^2}{L^2}\right).$$

**Corollary 1.** *Suppose* $\mathbf{u} \in \mathbb{R}^d$ *is a unit vector, and* $\mathbf{x} \sim \text{UNIFORM}\left(\mathbb{S}^{d-1}\right)$. *Then, for some constant* $C > 0$, *we have:*

$$\mathbb{P}\left(|\langle \mathbf{u}, \mathbf{x} \rangle| > \varepsilon\right) \le 2\exp\left(-Cd\varepsilon^2\right).$$

**Lemma 3.** *Suppose* $\widehat{\phi}_{t,1}^2, \ldots, \widehat{\phi}_{t,n_t}^2$ *are i.i.d. Bernoulli random variables with support* $\{\beta_t^2, (1-\beta_t)^2\}$, *where* $\beta_t > \frac{1}{2}$. *If* $\widehat{\phi}_t$ *is an i.i.d. copy of* $\widehat{\phi}_{t,i}$, *then the following inequality holds:*

$$\mathbb{P}\left(\left|\frac{1}{n_t}\sum_{i=1}^{n_t}\widehat{\phi}_{t,i}^2 - \mathbb{E}[\widehat{\phi}_t^2]\right| > \varepsilon\right) \le 2\exp\left(-\frac{2n_t\varepsilon^2}{(2\beta_t - 1)^2}\right).$$

**Proof of Lem. 3:** The result follows by applying the Chernoff bound for binomial distributions (Chernoff, 1952; Harchol-Balter, 2023) to the transformed random variables

$$\widetilde{\phi}_i := \frac{\widehat{\phi}_{t,i}^2 - (1-\beta_t)^2}{\beta_t^2 - (1-\beta_t)^2},$$

which are i.i.d. Bernoulli random variables taking values in $\{0, 1\}$.

**Lemma 4.** *Given the local linearity around* $\mathbf{x}_t$, *for any* $\omega_t \in \left[0, \frac{\delta}{\cos\alpha_t}\right]$, *we have:*

$$\mathbb{E}\left[\widehat{\phi}_t \langle \mathbf{g}_t, \mathbf{u} \rangle\right] = \frac{\Gamma\left(\frac{d}{2}\right) \cdot \left(1 - (\cos\alpha_t \cdot \omega_t/\delta)^2\right)^{\frac{d-1}{2}}}{2\sqrt{\pi} \cdot \Gamma\left(\frac{d+1}{2}\right)}$$

**Proof of Lem. 4:** By the law of total expectation and linearity assumption, we have

$$\mathbb{E}\left[\widehat{\phi}_t \langle \mathbf{g}_t, \mathbf{u} \rangle\right] = \beta_t \left(1 - P_t\left(\omega_t\right)\right) \mathbb{E}\left[-\langle \mathbf{g}_t, \mathbf{u} \rangle | \langle \mathbf{g}_t, \mathbf{u} \rangle \le -\cos\alpha_t \cdot \omega_t/\delta\right] \tag{15}$$
$$+ \left(1 - \beta_t\right) P_t\left(\omega_t\right) \mathbb{E}\left[\langle \mathbf{g}_t, \mathbf{u} \rangle | \langle \mathbf{g}_t, \mathbf{u} \rangle > -\cos\alpha_t \cdot \omega_t/\delta\right]$$

Now, by applying the divergence theorem on the constant vector field $\vec{\mathbf{F}} = \mathbf{g}_t$, we have

$$\mathbb{E}\left[-\langle \mathbf{g}_t, \mathbf{u} \rangle | \langle \mathbf{g}_t, \mathbf{u} \rangle \le -\cos\alpha_t \cdot \omega_t/\delta\right] = \left\langle -\mathbf{g}_t, \int_{\langle \mathbf{g}_t, \mathbf{u} \rangle \le -\cos\alpha_t \cdot \omega_t/\delta} \mathbf{u} p\left(\mathbf{u}\right) d\mathbf{u}\right\rangle$$

$$= \left\langle -\mathbf{g}_t, \int_{\langle \mathbf{g}_t, \mathbf{u} \rangle \le -\cos\alpha_t \cdot \omega_t/\delta} \frac{1}{\left(1 - p\left(\omega_t\right)\right) A_{d-1}\left(1\right)} \mathbf{u} d\mathbf{u}\right\rangle$$

$$= \frac{V_{d-1}\left(\sqrt{1 - (\cos\alpha_t \cdot \omega_t/\delta)^2}\right)}{\left(1 - P_t\left(\omega_t\right)\right) \cdot A_{d-1}\left(1\right)}$$

$$= \frac{\Gamma\left(\frac{d}{2}\right) \cdot \left(1 - (\cos\alpha_t \cdot \omega_t/\delta)^2\right)^{\frac{d-1}{2}}}{2\sqrt{\pi} \cdot \Gamma\left(\frac{d+1}{2}\right) \cdot \left(1 - P_t\left(\omega_t\right)\right)} \tag{16}$$

Where $V_d(r)$ and $A_{d-1}(r)$ are the volume of a $d$-dimensional ball and the area of a $d-1$-dimensional sphere with radius $r$, respectively. Note that the last equality comes from $V_d(r) = \dfrac{\pi^{d/2}}{\Gamma\left(\frac{d}{2} + 1\right)} r^d$ and $A_{d-1}(r) = \dfrac{2\pi^{d/2}}{\Gamma\left(\frac{d}{2}\right)} r^{d-1}$. Similarly, the following holds:

$$\mathbb{E}\left[\langle \mathbf{g}_t, \mathbf{u} \rangle | \langle \mathbf{g}_t, \mathbf{u} \rangle \ge -\cos\alpha_t \cdot \omega_t/\delta\right] = \frac{\Gamma\left(\frac{d}{2}\right) \cdot \left(1 - (\cos\alpha_t \cdot \omega_t/\delta)^2\right)^{\frac{d-1}{2}}}{2\sqrt{\pi} \cdot \Gamma\left(\frac{d+1}{2}\right) \cdot P_t\left(\omega_t\right)} \tag{17}$$

By using Eq. (16) and Eq. (17) in Eq. (15), we have

$$\mathbb{E}\left[\widehat{\phi}_t \langle \mathbf{g}_t, \mathbf{u} \rangle\right] = \frac{\Gamma\left(\frac{d}{2}\right) \cdot \left(1 - (\cos\alpha_t \cdot \omega_t/\delta)^2\right)^{\frac{d-1}{2}}}{2\sqrt{\pi} \cdot \Gamma\left(\frac{d+1}{2}\right)} \tag{18}$$

$\blacksquare$

**Lemma 5.** *Let $\varepsilon_1, \varepsilon_2 > 0$ be given. Then the following upper and lower bounds hold for $\mu(\mathbf{x}_t, \omega_t, \beta_t, n_t)$:*

*1. (Upper bound)*

$$\mu(\mathbf{x}_t, \omega_t, \beta_t, n_t) \leq \frac{\sqrt{n_t}\, \mathbb{E}\left[\widehat{\phi}_t \langle \mathbf{g}_t, \mathbf{u} \rangle\right]}{\sqrt{\mathbb{E}[\widehat{\phi}_t^2] - \varepsilon_2 - (n_t - 1)\beta_t^2 \varepsilon_1}}$$
$$+ \left(1 + \frac{\sqrt{n_t}\, \beta_t}{\sqrt{\mathbb{E}[\widehat{\phi}_t^2] - \varepsilon_2 - (n_t - 1)\beta_t^2 \varepsilon_1}}\right) K_{n_t, d}\left(\varepsilon_1, \varepsilon_2\right).$$

*2. (Lower bound)*

$$\mu(\mathbf{x}_t, \omega_t, \beta_t, n_t) \geq \frac{\sqrt{n_t}\, \mathbb{E}\left[\widehat{\phi}_t \langle \mathbf{g}_t, \mathbf{u} \rangle\right]}{\sqrt{\mathbb{E}[\widehat{\phi}_t^2] + \varepsilon_2 + (n_t - 1)\beta_t^2 \varepsilon_1}}$$
$$- \left(1 + \frac{\sqrt{n_t}\, \beta_t}{\sqrt{\mathbb{E}[\widehat{\phi}_t^2] + \varepsilon_2 + (n_t - 1)\beta_t^2 \varepsilon_1}}\right) K_{n_t, d}\left(\varepsilon_1, \varepsilon_2\right).$$

*Here, the error term $K_{n_t, d}(\varepsilon_1, \varepsilon_2)$ is defined as*

$$K_{n_t, d}\left(\varepsilon_1, \varepsilon_2\right) = n_t(n_t + 1) \exp\left(-Cd\varepsilon_1^2\right) + 2 \exp\left(-\frac{2n_t\varepsilon_2^2}{(2\beta_t - 1)^2}\right),$$

*for some universal constant $C > 0$.*

**Proof of Lem. 5:** Let $\varepsilon_1, \varepsilon_2 > 0$ be arbitrary. We define the following sets:
$$\mathcal{S}_i(\varepsilon_1) \coloneqq \{\mathbf{U} = (\mathbf{u}_1, \ldots, \mathbf{u}_{n_t}) \mid |\langle \mathbf{g}_t, \mathbf{u}_i \rangle| \geq \varepsilon_1\},$$
$$\mathcal{S}_{i,j}(\varepsilon_1) \coloneqq \{\mathbf{U} = (\mathbf{u}_1, \ldots, \mathbf{u}_{n_t}) \mid |\langle \mathbf{u}_i, \mathbf{u}_j \rangle| \geq \varepsilon_1\},$$
$$\mathcal{S}_\phi(\varepsilon_2) \coloneqq \left\{\mathbf{U} = (\mathbf{u}_1, \ldots, \mathbf{u}_{n_t}) \,\middle|\, \left|\mathbb{E}[\widehat{\phi}_t^2] - \frac{1}{n_t}\sum_{i=1}^{n_t} \widehat{\phi}_{t,i}^2\right| \geq \varepsilon_2\right\},$$
$$\mathcal{S}(\varepsilon_1, \varepsilon_2) \coloneqq \left(\bigcup_{i=1}^{n_t} \mathcal{S}_i(\varepsilon_1)\right) \cup \left(\bigcup_{1 \leq i < j \leq n_t} \mathcal{S}_{i,j}(\varepsilon_1)\right) \cup \mathcal{S}_\phi(\varepsilon_2).$$

For notational convenience, we also define
$$A(\mathbf{U}) \coloneqq \sum_{i=1}^{n_t} \widehat{\phi}_{t,i} \langle \mathbf{g}_t, \delta \mathbf{u}_i \rangle.$$

Applying Lem. 2 and Lem. 3, and using the union bound, we obtain

$$\mathbb{P}\left[\mathbf{U} \in \mathcal{S}\right] \leq n_t \cdot \mathbb{P}\left[\mathbf{U} \in \mathcal{S}_1\right] + \binom{n_t}{2} \cdot \mathbb{P}\left[\mathbf{U} \in \mathcal{S}_{1,2}\right] + \mathbb{P}\left[\mathbf{U} \in \mathcal{S}_\phi\right] \leq K_{n_t, d}\left(\varepsilon_1, \varepsilon_2\right). \tag{19}$$

Now, we derive the upper bound. By the law of total probability, we have

$$\mu(\mathbf{x}_t, \omega_t, \beta_t, n_t) = \mathbb{E}\left[\frac{\langle \mathbf{g}_t, \widehat{\nabla S}(\mathbf{x}_t, \omega_t, \beta_t) \rangle}{\left\|\widehat{\nabla S}(\mathbf{x}_t, \omega_t, \beta_t)\right\|_2}\right]$$
$$= \mathbb{E}\left[\frac{\langle \mathbf{g}_t, \widehat{\nabla S}(\mathbf{x}_t, \omega_t, \beta_t) \rangle}{\left\|\widehat{\nabla S}(\mathbf{x}_t, \omega_t, \beta_t)\right\|_2} \,\middle|\, \mathbf{U} \notin \mathcal{S}\right] \mathbb{P}[\mathbf{U} \notin \mathcal{S}]$$
$$+ \mathbb{E}\left[\frac{\langle \mathbf{g}_t, \widehat{\nabla S}(\mathbf{x}_t, \omega_t, \beta_t) \rangle}{\left\|\widehat{\nabla S}(\mathbf{x}_t, \omega_t, \beta_t)\right\|_2} \,\middle|\, \mathbf{U} \in \mathcal{S}\right] \mathbb{P}[\mathbf{U} \in \mathcal{S}].$$

To calculate the desired expected value given the event $\{\mathbf{U} \notin \mathcal{S}\}$, we expand the squared norm directly. By definition,

$$
\begin{aligned}
\left\| \widehat{\nabla S}(\mathbf{x}_t, \omega_t, \beta_t) \right\|_2^2 &= \left\| \frac{1}{n_t} \sum_{i=1}^{n_t} \delta \widehat{\phi}_{t,i} \mathbf{u}_i \right\|_2^2 \\
&= \frac{1}{n_t^2} \left\langle \sum_{i=1}^{n_t} \delta \widehat{\phi}_{t,i} \mathbf{u}_i, \sum_{j=1}^{n_t} \delta \widehat{\phi}_{t,j} \mathbf{u}_j \right\rangle \\
&= \frac{\delta^2}{n_t^2} \sum_{i=1}^{n_t} \sum_{j=1}^{n_t} \widehat{\phi}_{t,i} \widehat{\phi}_{t,j} \langle \mathbf{u}_i, \mathbf{u}_j \rangle \\
&= \frac{\delta^2}{n_t^2} \left( \sum_{i=1}^{n_t} \widehat{\phi}_{t,i}^2 + \sum_{\substack{1 \le i,j \le n_t \\ i \neq j}} \widehat{\phi}_{t,i} \widehat{\phi}_{t,j} \langle \mathbf{u}_i, \mathbf{u}_j \rangle \right).
\end{aligned}
$$

From the previous expansion, we have

$$
\left\| \widehat{\nabla S}(\mathbf{x}_t, \omega_t, \beta_t) \right\|_2 \ge \frac{\delta}{\sqrt{n_t}} \sqrt{\mathbb{E}[\widehat{\phi}_t^2] - \varepsilon_2 - (n_t - 1)\beta_t^2 \varepsilon_1}.
$$

Thus, on the event $\{\mathbf{U} \notin \mathcal{S}\}$, we can bound

$$
\frac{\langle \mathbf{g}_t, \widehat{\nabla S}(\mathbf{x}_t, \omega_t, \beta_t) \rangle}{\left\| \widehat{\nabla S}(\mathbf{x}_t, \omega_t, \beta_t) \right\|_2} = \frac{1}{\left\| \widehat{\nabla S} \right\|_2} \left\langle \mathbf{g}_t, \frac{1}{n_t} \sum_{i=1}^{n_t} \delta \widehat{\phi}_{t,i} \mathbf{u}_i \right\rangle = \frac{1}{n_t \left\| \widehat{\nabla S} \right\|_2} A(\mathbf{U}).
$$

Therefore,

$$
\frac{\langle \mathbf{g}_t, \widehat{\nabla S} \rangle}{\left\| \widehat{\nabla S} \right\|_2} \le \frac{A(\mathbf{U})}{\delta \sqrt{n_t} \sqrt{\mathbb{E}[\widehat{\phi}_t^2] - \varepsilon_2 - (n_t - 1)\beta_t^2 \varepsilon_1}}.
$$

Taking expectations, we get

$$
\mathbb{E}\left[ \frac{\langle \mathbf{g}_t, \widehat{\nabla S} \rangle}{\left\| \widehat{\nabla S} \right\|_2} \;\middle|\; \mathbf{U} \notin \mathcal{S} \right] \le \frac{1}{\delta \sqrt{n_t} \sqrt{\mathbb{E}[\widehat{\phi}_t^2] - \varepsilon_2 - (n_t - 1)\beta_t^2 \varepsilon_1}} \mathbb{E}[A(\mathbf{U})].
$$

Moreover, by independence and identical distribution of the samples, we have

$$
\mathbb{E}[A(\mathbf{U})] = n_t \delta \, \mathbb{E}\left[ \widehat{\phi}_t \langle \mathbf{g}_t, \mathbf{u} \rangle \right].
$$

Therefore,

$$
\mathbb{E}\left[ \frac{\langle \mathbf{g}_t, \widehat{\nabla S} \rangle}{\left\| \widehat{\nabla S} \right\|_2} \;\middle|\; \mathbf{U} \notin \mathcal{S} \right] \le \frac{\sqrt{n_t} \, \mathbb{E}\left[ \widehat{\phi}_t \langle \mathbf{g}_t, \mathbf{u} \rangle \right]}{\sqrt{\mathbb{E}[\widehat{\phi}_t^2] - \varepsilon_2 - (n_t - 1)\beta_t^2 \varepsilon_1}}.
$$

For the event $\{\mathbf{U} \in \mathcal{S}\}$, we use the trivial bound

$$
\left| \frac{\langle \mathbf{g}_t, \widehat{\nabla S} \rangle}{\left\| \widehat{\nabla S} \right\|_2} \right| \le 1,
$$

and hence

$$
\mathbb{E}\left[ \frac{\langle \mathbf{g}_t, \widehat{\nabla S} \rangle}{\left\| \widehat{\nabla S} \right\|_2} \;\middle|\; \mathbf{U} \in \mathcal{S} \right] \le 1.
$$

Substituting back into the law of total probability, we have

$$\mu(\mathbf{x}_t, \omega_t, \beta_t, n_t) \leq \frac{\sqrt{n_t}\, \mathbb{E}\left[\widehat{\phi}_t \langle \mathbf{g}_t, \mathbf{u}\rangle\right]}{\sqrt{\mathbb{E}[\widehat{\phi}_t^2] - \varepsilon_2 - (n_t - 1)\beta_t^2 \varepsilon_1}} (1 - \mathbb{P}[\mathbf{U} \in \mathcal{S}]) + \mathbb{P}[\mathbf{U} \in \mathcal{S}]$$
$$+ \frac{\sqrt{n_t}\beta_t \mathbb{P}[\mathbf{U} \in \mathcal{S}]}{\sqrt{\mathbb{E}[\widehat{\phi}_t^2] - \varepsilon_2 - (n_t - 1)\beta_t^2 \varepsilon_1}}.$$

Grouping terms, we obtain

$$\mu(\mathbf{x}_t, \omega_t, \beta_t, n_t) \leq \frac{\sqrt{n_t}\, \mathbb{E}\left[\widehat{\phi}_t \langle \mathbf{g}_t, \mathbf{u}\rangle\right]}{\sqrt{\mathbb{E}\left[\widehat{\phi}_t^2\right] - \varepsilon_2 - (n_t - 1)\beta_t^2 \varepsilon_1}}$$
$$+ \left(1 + \frac{\sqrt{n_t}\,\beta_t}{\sqrt{\mathbb{E}\left[\widehat{\phi}_t^2\right] - \varepsilon_2 - (n_t - 1)\beta_t^2 \varepsilon_1}}\right) \mathbb{P}\left[\mathbf{U} \in \mathcal{S}\right].$$

Applying the bound $\mathbb{P}\left[\mathbf{U} \in \mathcal{S}\right] \leq K_{n_t, d}(\varepsilon_1, \varepsilon_2)$ from Eq. (19), we finally get

$$\mu(\mathbf{x}_t, \omega_t, \beta_t, n_t) \leq \frac{\sqrt{n_t}\, \mathbb{E}\left[\widehat{\phi}_t \langle \mathbf{g}_t, \mathbf{u}\rangle\right]}{\sqrt{\mathbb{E}\left[\widehat{\phi}_t^2\right] - \varepsilon_2 - (n_t - 1)\beta_t^2 \varepsilon_1}}$$
$$+ \left(1 + \frac{\sqrt{n_t}\,\beta_t}{\sqrt{\mathbb{E}\left[\widehat{\phi}_t^2\right] - \varepsilon_2 - (n_t - 1)\beta_t^2 \varepsilon_1}}\right) K_{n_t, d}(\varepsilon_1, \varepsilon_2).$$

Similarly, for the lower bound, we have

$$\mu(\mathbf{x}_t, \omega_t, \beta_t, n_t) \geq \frac{\sqrt{n_t}\, \mathbb{E}\left[\widehat{\phi}_t \langle \mathbf{g}_t, \mathbf{u}\rangle\right]}{\sqrt{\mathbb{E}\left[\widehat{\phi}_t^2\right] + \varepsilon_2 + (n_t - 1)\beta_t^2 \varepsilon_1}}$$
$$- \left(1 + \frac{\sqrt{n_t}\,\beta_t}{\sqrt{\mathbb{E}\left[\widehat{\phi}_t^2\right] + \varepsilon_2 + (n_t - 1)\beta_t^2 \varepsilon_1}}\right) K_{n_t, d}(\varepsilon_1, \varepsilon_2).$$

This completes the proof of the lemma. We note that the argument does not rely on the linearity assumption.

∎

**Lemma 6.** *Assume local linearity holds around $\mathbf{x}_t$. Then, for any constant $q \in \left[\frac{1}{2}, 1\right)$, we have*

$$\mathbb{E}\left[\widehat{\phi}\left(\mathbf{x}_t + P_t^{-1}(q)\frac{\mathbf{x}_t - \mathbf{x}^\star}{\|\mathbf{x}_t - \mathbf{x}^\star\|_2} + \delta\mathbf{u}\right) \langle \mathbf{g}_t, \mathbf{u}\rangle\right] = \Theta\left(\frac{1}{\sqrt{d}}\right).$$

**Proof of Lem. 6:** Based on Lem. 4, we analyze the asymptotic behavior of $\frac{\Gamma\left(\frac{d}{2}\right)}{\Gamma\left(\frac{d+1}{2}\right)}$ and $\left(1 - (\cos\alpha_t \cdot \omega_t/\delta)^2\right)^{\frac{d-1}{2}}$ as $d$ tends to infinity.

By Lem. 2, we have

$$\mathbb{P}\left[|\langle \mathbf{u}, \mathbf{g}_t\rangle| > \varepsilon\right] \leq 2\exp\left(-Cd\varepsilon^2\right),$$

so in particular

$$\mathbb{P}\left[\langle \mathbf{u}, \mathbf{g}_t \rangle < -\varepsilon\right] \leq \exp\left(-Cd\varepsilon^2\right).$$

We know that $1 - q = \mathbb{P}\left[\langle \mathbf{u}, \mathbf{g}_t \rangle > -\cos\alpha_t \cdot \omega_t/\delta\right]$, based on the selection of the overshooting value $\omega_t$. Thus,

$$1 - q \leq \exp\left(-Cd\left(\cos\alpha_t \cdot \omega_t/\delta\right)^2\right),$$

which implies

$$\ln(1 - q) \leq -Cd\left(\cos\alpha_t \cdot \omega_t/\delta\right)^2,$$

and consequently

$$1 + \frac{\ln(1-q)}{Cd} \leq 1 - \left(\cos\alpha_t \cdot \omega_t/\delta\right)^2.$$

Raising both sides to the $(d-1)/2$ power yields

$$\left(1 + \frac{\ln(1-q)}{Cd}\right)^{\frac{d-1}{2}} \leq \left(1 - \left(\cos\alpha_t \cdot \omega_t/\delta\right)^2\right)^{\frac{d-1}{2}}.$$

Applying the classical limit $\lim_{n\to\infty}\left(1 + \frac{c}{n}\right)^n = e^c$, we obtain

$$\lim_{d\to\infty}\left(1 + \frac{\ln(1-q)}{Cd}\right)^{\frac{d-1}{2}} = \exp\left(\frac{\ln(1-q)}{2C}\right),$$

which implies that $\left(1 - \left(\cos\alpha_t \cdot \omega_t/\delta\right)^2\right)^{\frac{d-1}{2}} = \Theta(1)$.

On the other hand, by Stirling's approximation

$$\Gamma(n) = \sqrt{\frac{2\pi}{n}}\left(\frac{n}{e}\right)^n\left(1 + \mathcal{O}\left(\frac{1}{n}\right)\right),$$

we find that

$$\frac{\Gamma\left(\frac{d}{2}\right)}{\Gamma\left(\frac{d+1}{2}\right)} = \Theta\left(\frac{1}{\sqrt{d}}\right).$$

Substituting these results into Eq. (18) concludes the proof.

∎

Now, we proceed to prove the theorem.

**Proof of Thm. 2:** For any $0 < z < \frac{1}{8}$, let $n_t = d^{3z}$, $\varepsilon_1 = d^{-4z}$, and $\varepsilon_2 = d^{-z}$. Also, let $d \geq 4\frac{\mathbb{E}\left[\widehat{\phi}_t^2\right]}{\beta^2}$. We define

$$E_1 := \frac{\sqrt{n_t}\,\mathbb{E}\left[\widehat{\phi}_t\langle \mathbf{g}_t, \mathbf{u}\rangle\right]}{\sqrt{\mathbb{E}\left[\widehat{\phi}_t^2\right] - \varepsilon_2 - (n_t - 1)\beta^2\varepsilon_1}},$$

$$E_2 := \frac{\sqrt{n_t}\,\beta}{\sqrt{\mathbb{E}\left[\widehat{\phi}_t^2\right] - \varepsilon_2 - (n_t - 1)\beta^2\varepsilon_1}},$$

$$E_3 := K_{n_t,d}(\varepsilon_1, \varepsilon_2).$$

Then using the upper bound derived in Lem. 5, we have

$$\frac{\mu(\mathbf{x}_t, p_t^{-1}(q), \beta, n_t)}{J(\mathbf{x}_t, p_t^{-1}(q), \beta, n_t)} - 1 \leq \frac{E_1}{J(\mathbf{x}_t, p_t^{-1}(q), \beta, n_t)} - 1 + \frac{1 + E_2}{J(\mathbf{x}_t, p_t^{-1}(q), \beta, n_t)} \cdot E_3. \tag{20}$$

Since $\mathbb{E}\left[\widehat{\phi}_t^2\right] = \beta^2(1-q) + (1-\beta)^2 q$ is constant, we can estimate

$$\frac{E_1}{J(\mathbf{x}_t, p_t^{-1}(q), \beta, n_t)} - 1 \le \frac{1}{\sqrt{1 - \frac{\varepsilon_2 + (n_t - 1)\beta^2 \varepsilon_1}{\mathbb{E}[\widehat{\phi}_t^2]}}} - 1$$

$$\le \mathcal{O}\left(\varepsilon_2 + (n_t - 1)\beta^2 \varepsilon_1\right) \quad \text{(since for } 0 \le x \le \frac{1}{2}, \ \sqrt{1-x} \ge 1 - \frac{x}{2})$$

$$= \mathcal{O}(d^{-z}). \tag{21}$$

Moreover, we have $E_1 = \Theta\left(n_t^{1/2}\right)$. Using Lem. 6, we also have $J(\mathbf{x}_t, p_t^{-1}(q), \beta, n_t) = \Theta\left(n_t^{1/2} d^{1/2}\right)$. Thus,

$$\frac{1 + E_2}{J(\mathbf{x}_t, p_t^{-1}(q), \beta, n_t)} = \Theta(d^{1/2}). \tag{22}$$

Substituting the values of $n_t$, $\varepsilon_1$, and $\varepsilon_2$ into $E_3$ yields

$$E_3 = \Theta\left(d^{6z}\right) \exp\left(-Cd^{1-8z}\right) + \exp\left(-\frac{2d^z}{(2\beta - 1)^2}\right). \tag{23}$$

Since $1 - 8z > 0$ and exponential functions dominate polynomial growth, combining Eq. (21), Eq. (22), and Eq. (23) with Eq. (20) yields

$$\frac{\mu(\mathbf{x}_t, p_t^{-1}(q), \beta, n_t)}{J(\mathbf{x}_t, p_t^{-1}(q), \beta, n_t)} - 1 \le \mathcal{O}(d^{-z}).$$

Applying similar steps using the lower bound in Lem. 5, we find

$$1 - \frac{\mu(\mathbf{x}_t, p_t^{-1}(q), \beta, n_t)}{J(\mathbf{x}_t, p_t^{-1}(q), \beta, n_t)} \le \mathcal{O}(d^{-z}).$$

Thus, the proof is complete.

∎

### B.3 PROOF OF THM. 3

Since $J(\mathbf{x}_t, \omega_t, \beta_t, n_t)$ is increasing with respect to $n_t$, the optimal choice is to take $n_t$ at its maximum allowed value:

$$n_t = \frac{c_t}{c^\star - (c^\star - 1)P_t(\omega_t^\star)}.$$

Substituting this into the definition of $J$ and applying Lem. 4, we obtain

$$J(\mathbf{x}_t, \omega_t, \beta_t, n_t) \propto \frac{(1 - (\cos\alpha_t \cdot \omega_t/\delta)^2)^{(d-1)/2}}{\sqrt{(c^\star - (c^\star - 1)P_t(\omega_t))\, \mathbb{E}[\widehat{\phi}_t^2]}}.$$

Expanding $\mathbb{E}[\widehat{\phi}_t^2]$ gives

$$\mathbb{E}[\widehat{\phi}_t^2] = \beta_t^2(1 - P_t(\omega_t)) + (1 - \beta_t)^2 P_t(\omega_t).$$

Thus,

$$J(\mathbf{x}_t, \omega_t, \beta_t, n_t) \propto \frac{(1 - (\cos\alpha_t \omega_t/\delta)^2)^{(d-1)/2}}{\sqrt{(c^\star - (c^\star - 1)P_t(\omega_t))(\beta_t^2(1 - P_t(\omega_t)) + (1 - \beta_t)^2 P_t(\omega_t))}}.$$

To maximize $J$, it suffices to minimize

$$\beta_t^2(1 - P_t(\omega_t)) + (1 - \beta_t)^2 P_t(\omega_t).$$

Differentiating with respect to $\beta_t$ and setting the derivative to zero yields

$$\beta_t = P_t(\omega_t).$$

Substituting this optimal $\beta_t$ back, we find

$$\mathbb{E}[\widehat{\phi}_t^2] = P_t(\omega_t)(1 - P_t(\omega_t)),$$

and thus the final expression to maximize is

$$\widehat{J}_t(\omega_t) = \frac{(1 - (\cos\alpha_t \omega_t / \delta)^2)^{d-1}}{P_t(\omega_t)(1 - P_t(\omega_t))(c^\star - (c^\star - 1)P_t(\omega_t))}.$$

∎

### B.4 PROOF OF THM. 4

First, we show that $\cos\alpha_1 \geq 0$. Suppose for contradiction that $\cos\alpha_1 < 0$, i.e.,

$$\langle \mathbf{x}_1 - \mathbf{x}^\star, \nabla S(\mathbf{x}_1) \rangle < 0.$$

Using the definition of the directional derivative, we have

$$\lim_{h \to 0} \frac{S(\mathbf{x}_1 + h(\mathbf{x}_1 - \mathbf{x}^\star)) - S(\mathbf{x}_1)}{h} = \langle \mathbf{x}_1 - \mathbf{x}^\star, \nabla S(\mathbf{x}_1) \rangle < 0.$$

Since the directional derivative is strictly negative, there exists $\epsilon > 0$ such that for all sufficiently small $\epsilon > 0$,

$$S(\mathbf{x}_1 - \epsilon(\mathbf{x}_1 - \mathbf{x}^\star)) > S(\mathbf{x}_1).$$

Noting that

$$\mathbf{x}_1 - \epsilon(\mathbf{x}_1 - \mathbf{x}^\star) = \mathbf{x}^\star + (1 - \epsilon)(\mathbf{x}_1 - \mathbf{x}^\star),$$

we can rewrite this inequality as

$$S(\mathbf{x}^\star + (1 - \epsilon)(\mathbf{x}_1 - \mathbf{x}^\star)) > S(\mathbf{x}_1).$$

Since $\phi(\mathbf{x}_1) = 1$ by assumption, and assuming $\phi$ remains 1 in a neighborhood where $S$ does not decrease, we also have

$$\phi(\mathbf{x}^\star + (1 - \epsilon)(\mathbf{x}_1 - \mathbf{x}^\star)) = 1.$$

Thus, for $r = 1 - \epsilon$, we find a point $0 < r < 1$ such that $\phi(\mathbf{x}^\star + r(\mathbf{x}_1 - \mathbf{x}^\star)) = 1$, contradicting the assumption that no such $r$ exists. Therefore, our assumption that $\cos\alpha_1 < 0$ must be false, and we conclude that

$$\cos\alpha_1 \geq 0.$$

Now that we have established $\cos\alpha_1 \geq 0$, it follows that

$$\mathbb{E}[\cos\alpha_1] = \mathbb{E}[\cos\alpha_1 \mid \cos\alpha_1 \geq 0].$$

Expanding $\cos\alpha_1$ in terms of the vectors involved, we write

$$\cos\alpha_1 = \left\langle \frac{\mathbf{x}_1 - \mathbf{x}^\star}{\|\mathbf{x}_1 - \mathbf{x}^\star\|_2}, \mathbf{g}_1 \right\rangle.$$

Thus,

$$\mathbb{E}[\cos\alpha_1] = \mathbb{E}\left[\left\langle \frac{\mathbf{x}_1 - \mathbf{x}^\star}{\|\mathbf{x}_1 - \mathbf{x}^\star\|_2}, \mathbf{g}_1 \right\rangle \;\middle|\; \left\langle \frac{\mathbf{x}_1 - \mathbf{x}^\star}{\|\mathbf{x}_1 - \mathbf{x}^\star\|_2}, \mathbf{g}_1 \right\rangle \geq 0 \right].$$

Finally, applying the result from Eq. (17) with $\omega_t = 0$, we obtain

$$\mathbb{E}\left[\left\langle \frac{\mathbf{x}_1 - \mathbf{x}^\star}{\|\mathbf{x}_1 - \mathbf{x}^\star\|_2}, \mathbf{g}_1 \right\rangle \;\middle|\; \left\langle \frac{\mathbf{x}_1 - \mathbf{x}^\star}{\|\mathbf{x}_1 - \mathbf{x}^\star\|_2}, \mathbf{g}_1 \right\rangle \geq 0 \right] = \frac{\Gamma\left(\frac{d}{2}\right)}{2\sqrt{\pi}\,\Gamma\left(\frac{d+1}{2}\right)}.$$

This completes the proof.

∎

## C EMPIRICAL STUDY OF AS AND AGREST

Here, we design two experiments to validate the effectiveness of our analysis for both AS and AGREST. (Fig. 4)

First, we sample 100 random correctly classified images from the ImageNet dataset and run the experiments using binary search and asymmetric search when $c^\star = 10^3$. We observe that the average cumulative search cost across iterations for binary search is approximately 2.5 times higher than that of AS. This highlights the effectiveness of AS compared to vanilla search.

Second, to show that using the overshooting value obtained by AGREST leads to a probability of making low-cost queries close to the theoretical value in Thm. 3, we again sample 100 random images and run one iteration of AGREST using 500 queries for gradient estimation. We then compute the empirical probability of making low-cost queries, defined as the ratio of low-cost to total (500) queries, and compare it to the optimal probability predicted by our theoretical analysis. As shown in Fig. 4, our analysis is close to the empirical results, especially for larger values of $c^\star$.

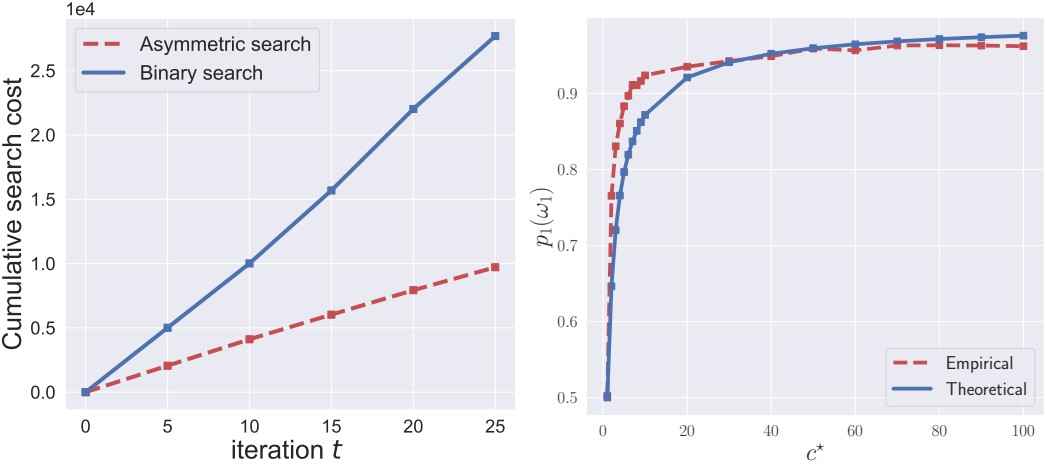

Figure 4: Empirical study of AS and AGREST. The left plot compares AS with vanilla search (binary search) in terms of cumulative *search* cost over iterations in GeoDA when $c^\star = 10^3$, while the right plot shows the optimal theoretical probability of making low-cost queries (assuming local linearity of the decision boundary) versus the empirical ratio of low-cost to total queries for different values of $c^\star$.

| $c^\star$ | $m = 0.02$ | $m = 0.04$ | $m = 0.06$ | $m = 0.08$ | $m = 0.10$ |
|---|---|---|---|---|---|
| 2 | 2.45 | 2.91 | 2.87 | 2.54 | 3.16 |
| 5 | 3.52 | 3.59 | 3.75 | 3.37 | 3.62 |
| 100 | 3.02 | 4.36 | 3.43 | 4.20 | 4.95 |
| 1000 | 17.69 | 19.71 | 23.75 | 22.84 | 30.10 |
| 10000 | 12.91 | 17.30 | 23.72 | 24.728 | 24.004 |

Table 2: Peformance of HSJA+AGREST to $m$ across different values of $c^\star$ (25 random images).

# D   IMPLEMENTATION

## D.1   MODIFICATION TO GEODA

We modify the direction-based adversarial example search phase in GeoDA. In its original implementation, GeoDA estimates the gradient and then proceeds from the original image, taking fixed-size steps along that direction until it finds an adversarial example. However, this process often leads to a large number of flagged queries, since many of the intermediate steps can cross the decision boundary.

To address this issue, we change the starting point of the search. Instead of beginning at the original image, we start from

$$\mathbf{x}'' = \mathbf{x}^\star + \|\mathbf{x}^\star - \mathbf{x}_t\|_2 \cdot \frac{\widehat{\nabla} S(\mathbf{x}_t, \omega_t)}{\|\widehat{\nabla} S(\mathbf{x}_t, \omega_t)\|_2}.$$

This new starting point lies further in the direction of the estimated gradient and is designed with the expectation that it is already adversarial—or at least closer to an adversarial example than the original image. If $\mathbf{x}''$ is not adversarial, the algorithm continues the search in the estimated gradient direction. This modification significantly reduces the number of flagged queries encountered during the search.

## D.2   SELECTION OF THE HYPERPARAMETER $m$

We select values for the hyperparameter $m$ by evaluating the performance of the corresponding attacks under different settings of $m$, using 20 randomly selected correctly classified images. This evaluation is performed with $c^\star = 10^3$ and a total query cost of 250K, as shown in Fig. 5.

Although the optimal value of $m$ can vary with $c^\star$, we choose to fix $m$ independently of $c^\star$. This decision simplifies the attack process and avoids the additional computational overhead of tuning $m$ for each value of $c^\star$, while still enabling effective attack performance.

**Sensitivity of the hyperparameter.**   We report in Tab. 2 the results of running HSJA with AGREST on 25 randomly selected images for different values of $c^\star$ and $m$, indicating that performance is only weakly sensitive to the hyperparameter $m$. This suggests that $m$ can be transferred from a known model for a fixed $c^\star$.

## D.3   AGREST WITH DIMENSION REDUCTION

As mentioned earlier, most practical attacks use a dimension reduction matrix $\mathbf{R} \in \mathbb{R}^{d \times d'}$ to perform the sampling process in a subspace of dimension $d' \ll d$, where $d$ is the dimension of the original space, in order to increase sample efficiency. To apply the same subspace in the AGREST estimator, the only modifications needed compared to the original AGREST Alg. 3 are: first, projecting each sample into the subspace; and second, using the effective dimension $d'$ to compute $\alpha_t$. An overview of this version of AGREST is provided in Alg. 4.

## D.4   COMPUTATION RESOURCES

For our experiments on ResNet-50, we use NVIDIA P100 GPUs. All other experiments, including those involving ViT and CLIP models, are conducted on NVIDIA A100 GPUs to accommodate the higher computational and memory demands of these models.

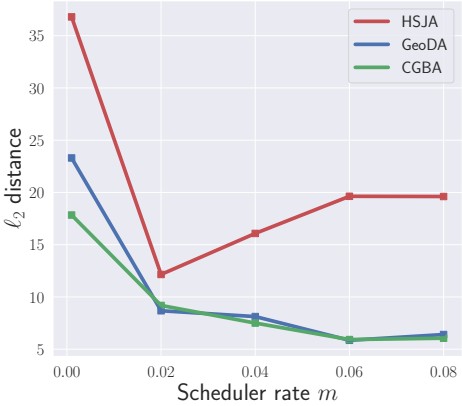

Figure 5: Median $\ell_2$ distance of adversarial perturbations for varying values of $m$, with $c^\star = 10^3$ and a total query cost of 150K.

---

**Algorithm 4** AGREST Estimation

---

**Inputs:** Iteration $t$, source image $\mathbf{x}^\star$, boundary point $\mathbf{x}_t$, dimension $d$, sampling subspace dimension $d'$, sampling subspace matrix $\mathbf{R}$, high-cost query cost $c^\star$, sampling radius $\delta$, sampling batch size $b$, cosine value $\alpha_t$, vanilla gradient estimation query budget $n_t'$, scheduler rate $m$

**Outputs:** Normalized approximated direction $g_t$, next cosine value $\alpha_{t+1}$

1: $n_L \leftarrow 0, \; n_H \leftarrow 0, \; \mathbf{v}^+ \leftarrow \vec{0}, \; \mathbf{v}^- \leftarrow \vec{0}, \; \hat{c} \leftarrow 0, \; \omega^\star \leftarrow \text{OVERSHOOTING}\,(c^\star)$      ▷ Thm. 3
2: $\omega_t \leftarrow \omega^\star / \cos\alpha_t, \; c_t \leftarrow n_t'(c^\star + 1)/2$
3: **while** $\hat{c} < c_t$ **do**
4:     $B \leftarrow \left\{ \mathbf{R}\mathbf{u}_i / \|\mathbf{R}\mathbf{u}_i\| \text{ where } \mathbf{u}_i \sim \text{UNIFORM}\left(\mathbb{S}^{d-1}\right) \right\}_{i=1}^{b}$      ▷ Dimension reduction
5:     **for each** $\mathbf{u}_i \in B$ **do**
6:        **if** $\phi\left(\mathbf{x}_t + \omega_t \frac{\mathbf{x}_t - \mathbf{x}^\star}{\|\mathbf{x}_t - \mathbf{x}^\star\|_2} + \delta\mathbf{u}_i\right) = 1$ **then**
7:          $\mathbf{v}^+ \leftarrow \mathbf{v}^+ + \mathbf{u}_i, \; n_L \leftarrow n_L + 1, \; \hat{c} \leftarrow \hat{c} + 1$
8:        **else**
9:          $\mathbf{v}^- \leftarrow \mathbf{v}^- - \mathbf{u}_i, \; n_H \leftarrow n_H + 1, \; \hat{c} \leftarrow \hat{c} + c^\star$
10:       **end if**
11:     **end for**
12: **end while**
13: $\hat{p}_t \leftarrow n_L / (n_L + n_H)$
14: $g_t \leftarrow (1 - \hat{p}_t)\,\mathbf{v}^+ + \hat{p}_t\mathbf{v}^-, \; \alpha_{t+1} \leftarrow \text{SCHEDULER-STEP}\,(t, \hat{p}_t, m)$      ▷ Alg. 2
15: **return** $g_t / \|g_t\|_2, \; \alpha_{t+1}$

---

## E   MORE RESULTS

### E.1   ASR COMPARISONS

In this section, we present a detailed comparison of attack success rates (ASR) achieved by various methods under perturbation norms $\ell_2 = 5$ and $\ell_2 = 10$ on ResNet-50. The following results highlight that our proposed enhancements consistently deliver substantial gains, reaching up to 20% improvement in ASR.

### E.2   QUALITATIVE

Fig. 6 presents qualitative results for different HSJA variants at $c^\star = 1000$. As can be seen, incorporating AS and AGREST improves the qualitative quality of the generated adversarial examples.

Table 3: ASR (%) under $\ell_2 = 5$ for different $(c^\star, \text{total cost})$ on ResNet-50.

| Attack | Variant | (2,10000) | (5,15000) | (100,150000) | (1000,250000) |
|--------|---------|-----------|-----------|--------------|---------------|
| SURFREE | VA | 55.4 | 48.7 | 37.2 | 16.7 |
| SURFREE | VA+AS | 63.9 | 63.9 | 57.8 | 37.0 |
| HSJA | VA | 74.2 | 69.8 | 52.8 | 18.5 |
| HSJA | VA+AS | 75.4 | 67.7 | 55.7 | 18.8 |
| HSJA | VA+AGREST | 77.7 | 73.0 | 73.4 | 22.6 |
| HSJA | VA+AGREST+AS | 76.8 | 73.6 | 74.8 | 24.2 |
| GEODA | VA | 66.3 | 61.6 | 56.0 | 29.9 |
| GEODA | VA+AS | 65.4 | 61.9 | 57.5 | 32.3 |
| GEODA | VA+AGREST | 61.6 | 65.7 | 73.3 | 43.4 |
| GEODA | VA+AGREST+AS | 60.4 | 66.0 | 69.3 | 46.1 |
| CGBA | VA | **92.7** | 88.0 | 78.0 | 29.9 |
| CGBA | VA+AS | 92.1 | 88.9 | 78.0 | 31.8 |
| CGBA | VA+AGREST | 91.8 | 89.4 | 89.5 | **45.5** |
| CGBA | VA+AGREST+AS | 91.5 | **89.7** | **90.6** | 43.0 |

Table 4: ASR (%) under $\ell_2 = 10$ for different $(c^\star, \text{total cost})$ on ResNet-50.

| Attack | Variant | (2,10000) | (5,15000) | (100,150000) | (1000,250000) |
|--------|---------|-----------|-----------|--------------|---------------|
| SURFREE | VA | 80.4 | 71.0 | 58.4 | 31.1 |
| SURFREE | VA+AS | 81.8 | 82.7 | 78.9 | 59.8 |
| HSJA | VA | 89.1 | 86.2 | 73.9 | 25.2 |
| HSJA | VA+AS | 90.9 | 85.9 | 76.5 | 29.6 |
| HSJA | VA+AGREST | 90.6 | 88.9 | 89.3 | 37.8 |
| HSJA | VA+AGREST+AS | 90.6 | 89.4 | 90.8 | 43.0 |
| GEODA | VA | 83.0 | 79.8 | 75.4 | 47.5 |
| GEODA | VA+AS | 81.2 | 77.7 | 75.4 | 53.1 |
| GEODA | VA+AGREST | 79.2 | 82.1 | 84.8 | 65.4 |
| GEODA | VA+AGREST+AS | 78.6 | 79.8 | 84.3 | 68.4 |
| CGBA | VA | **99.1** | 97.7 | 92.7 | 50.1 |
| CGBA | VA+AS | **99.1** | 97.7 | 93.0 | 52.1 |
| CGBA | VA+AGREST | **99.1** | **98.5** | 99.0 | **70.4** |
| CGBA | VA+AGREST+AS | 98.2 | **98.5** | **99.3** | 65.0 |

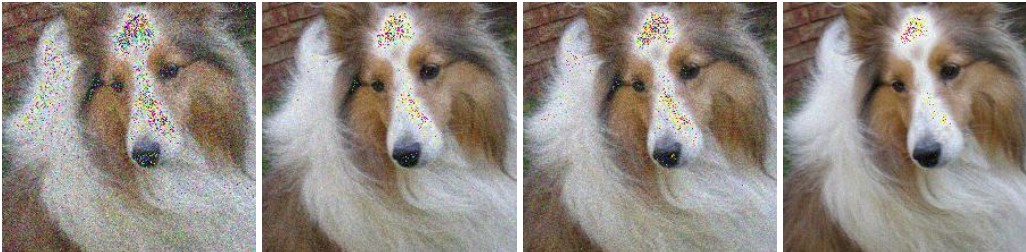

Figure 6: Qualitative results for HSJA variants at $c^\star$. From left to right: vanilla, AS, AGREST, and VA+AGREST.

Table 5: Transfer ASR (%) for different source and target models using PGD-40 with $\ell_2 = 10$.

| Source Model | Target Model | Transfer ASR |
|---|---|---|
| VGG19 | ResNet50 | 79.6 |
| VGG19 | ViT-B/32 | 9.4 |
| ViT-B/16 | ResNet50 | 41.0 |
| ViT-B/16 | ViT-B/32 | 28.4 |

### E.3 TRANSFER ATTACKS

We evaluated the transferability of adversarial examples generated from different surrogate models under a PGD-40 attack with an $\ell_2$ norm radius of 10. Tab. 5 summarizes the ASR when transferring from VGG19 and ViT-B/16 to ResNet50 and ViT-B/32.

Transfer attacks usually have lower success rates than query-heavy decision-based methods, but they are especially useful when the budget is limited. For example, transferring from VGG19 to ResNet50 achieves a 79.6% attack success rate without using any queries.

These results show the complementary role of transfer-based strategies. They may not always be strong across every model pair, but they provide a powerful and cost-free option when query budgets are very limited.

### E.4 ASYMMETRIC ATTACKS VS ADVERSARIAL DEFENSES

In this section, we evaluate VA and AGREST under several adversarial defenses. To this end, we use the PGD-trained model with $\ell_2 = 3$ (whose clean accuracy drops by roughly 20%) and observe that both VA and AGREST improve over the vanilla HSJA attack (Tab. 6).

| Method | $c^\star = 100$ | $c^\star = 1000$ |
|---|---|---|
| VA | 67.43 | 127.18 |
| VA+AS | 66.02 | 116.59 |
| VA+AGREST | 61.88 | 108.75 |
| VA+AS+AGREST | 60.39 | 85.85 |

Table 6: HSJA variant performance on the PGD-trained robust model ($\ell_2 = 3$) across different values of $c^\star$.

In addition to the white-box defense, we also evaluate quantization as a defense against black-box attacks, which results in a much smaller reduction in clean accuracy compared to white-box defenses. We observe further improvements when using VA and AGREST (Tab. 7).

### E.5 CIFAR-10 ON RESNET-20: RESULTS FOR HSJA

Our experiments on CIFAR-10, where the number of classes is significantly smaller than in ImageNet, indicate that the method continues to deliver substantial improvements. Even in this setting, our method achieves up to a 40% reduction in the perturbation norm (Tab. 8).

| Method | $c^\star = 100$ | $c^\star = 1000$ |
|---|---|---|
| VA | 9.13 | 37.60 |
| VA+AS | 8.82 | 36.93 |
| VA+AGREST | 6.70 | 34.79 |
| VA+AS+AGREST | 7.08 | 31.48 |

Table 7: HSJA variants under quantization defense.

| Variant | $(2, 10000)$ | $(5, 15000)$ | $(100, 150000)$ | $(1000, 250000)$ |
|---|---|---|---|---|
| VA | 0.130 | 0.148 | 0.186 | 0.584 |
| VA + AS | 0.129 | 0.147 | 0.184 | 0.506 |
| VA + AGREST | 0.132 | 0.142 | 0.137 | 0.409 |
| VA + AGREST + AS | 0.133 | 0.139 | 0.130 | 0.309 |

Table 8: Performance of HSJA variants. Each pair is reported in the form $(c^\star, \text{total\_cost})$.

# F  ADDITIONAL RESULTS FOR VISION TRANSFORMERS (ViT)

In this section, we present comprehensive empirical evaluations that extend our analysis across varying budget constraints and different query cost parameters $c^\star$. Specifically, we conduct experiments utilizing Vision Transformer architectures (ViT-B/32 and ViT-B/16) on the ImageNet dataset.

Table 9: Median $\ell_2$ distance for various $c^\star$ values and different types of attacks for **ViT-B/32** model on ImageNet dataset.

| Attack | Method | $c^\star = 2.0$ Total Cost | | | | | $c^\star = 5.0$ Total Cost | | | | | $c^\star = 100.0$ Total Cost | | | | | $c^\star = 1000.0$ Total Cost | | | | | Higher Queries Total Cost | |
|---|---|---|---|---|---|---|---|---|---|---|---|---|---|---|---|---|---|---|---|---|---|---|---|
| | | 1000 | 2000 | 5000 | 10000 | 15000 | 1000 | 2000 | 5000 | 10000 | 15000 | 1000 | 2000 | 5000 | 10000 | 15000 | 1000 | 2000 | 5000 | 10000 | 15000 | 150000 | 250000 |
| SURFREE | VA | 9.3 | 6.2 | 3.9 | 2.9 | 2.5 | 14.3 | 9.9 | 5.7 | 4.4 | 3.4 | 70.4 | 71.0 | 34.7 | 21.3 | 18.2 | **69.7** | 71.1 | **70.6** | 68.1 | **68.3** | 5.13 | 16.12 |
| | VA+AS (A-SurFree) | **7.8** | **5.0** | **3.6** | **2.4** | **2.1** | **11.4** | **7.3** | **4.6** | **3.1** | **2.5** | **70.0** | **70.1** | 22.8 | 14.5 | 9.7 | 70.1 | 69.9 | 71.5 | 67.3 | 68.5 | 3.68 | 6.35 |
| HSJA | VA | 53.7 | 44.8 | 28.7 | 18.3 | 13.9 | 61.6 | 53.2 | 40.0 | 28.1 | 22.8 | **68.4** | **67.2** | 67.5 | 67.9 | 63.1 | 69.9 | 71.1 | 69.3 | 69.4 | 70.2 | 4.21 | 22.46 |
| | VA+AS | 53.3 | 41.8 | 25.2 | 18.3 | 13.5 | 62.6 | 50.5 | 38.6 | 24.4 | 18.5 | 69.5 | 68.4 | 69.7 | 58.8 | 56.8 | 68.7 | **69.4** | 70.2 | 68.1 | **68.2** | 3.88 | 18.79 |
| | VA+AGREST | 18.6 | 9.7 | **4.2** | 2.7 | 2.2 | 36.3 | **17.3** | 7.8 | 4.4 | **3.1** | 69.1 | 71.1 | 73.7 | **37.0** | 35.4 | **68.5** | 72.1 | **69.1** | 70.1 | 69.7 | 2.19 | 11.28 |
| | VA+AS+AGREST (A-HSJA) | **17.8** | **9.6** | 4.7 | **2.7** | **2.1** | **34.8** | 19.1 | **7.6** | **4.2** | **3.1** | 71.4 | 69.1 | **60.1** | 38.3 | **25.3** | 71.8 | 72.1 | 71.9 | **67.6** | 68.7 | 2.06 | 10.74 |
| GEODA | VA | 14.2 | 6.6 | 3.4 | 2.7 | 2.4 | 18.9 | 12.0 | 6.7 | 3.5 | 3.2 | **67.1** | **68.2** | **30.1** | 26.3 | 18.7 | 69.3 | **68.6** | 68.1 | 67.5 | 69.2 | 3.97 | 9.83 |
| | VA+AS | 12.7 | 7.8 | 3.7 | 2.7 | 2.3 | 17.1 | 11.7 | 5.5 | 3.4 | 2.9 | 71.0 | 73.7 | 30.5 | 20.5 | 16.0 | 70.5 | 71.5 | 68.7 | 73.1 | 71.4 | 3.12 | 8.78 |
| | VA+AGREST | **7.4** | **3.7** | 2.4 | **1.9** | **1.7** | **14.6** | 8.8 | 4.5 | 3.3 | **2.7** | 70.2 | 69.8 | 36.6 | 23.5 | 17.7 | **68.6** | 72.2 | **66.0** | 69.0 | 69.5 | 2.10 | 5.12 |
| | VA+AS+AGREST (A-GeoDA) | 7.8 | 3.9 | **2.3** | **1.9** | 1.8 | 14.8 | **8.4** | **4.4** | **3.1** | 2.8 | 68.8 | 70.6 | 34.0 | **19.0** | **15.6** | 72.2 | 72.5 | 73.5 | 72.9 | **68.2** | 2.03 | 4.35 |
| CGBA | VA | 11.6 | 6.0 | 3.0 | 1.6 | 1.4 | 18.4 | 11.3 | 5.3 | 3.1 | 2.1 | 70.4 | 72.0 | 28.0 | 24.5 | 17.0 | **68.8** | 70.4 | 69.7 | 72.8 | 72.8 | 2.13 | 9.67 |
| | VA+AS | 10.8 | 6.4 | 3.2 | 1.7 | 1.3 | 16.2 | 10.7 | 4.6 | **2.6** | 2.0 | 68.5 | 73.4 | 29.2 | 18.1 | 13.7 | 71.9 | 73.8 | 68.3 | **68.5** | 70.1 | 1.97 | 8.24 |
| | VA+AGREST | 7.3 | 3.6 | 2.0 | 1.5 | 1.3 | 15.8 | 8.3 | **4.0** | 3.0 | 2.5 | **67.1** | 73.5 | 39.2 | 18.9 | 15.2 | 70.9 | 70.5 | 72.7 | 69.8 | **69.5** | 1.56 | 5.46 |
| | VA+AS+AGREST (A-CGBA) | 7.6 | 4.1 | **1.9** | **1.5** | **1.2** | **14.4** | **7.7** | 4.2 | 2.9 | 2.5 | 70.2 | **67.2** | **23.9** | **15.9** | **13.4** | 71.9 | 70.4 | **67.3** | 70.0 | 72.7 | 1.42 | 5.61 |

Table 10: Median $\ell_2$ distance for various $c^\star$ values and different types of attacks for **ViT-B/16** model on ImageNet dataset.

| Attack | Method | $c^\star = 2.0$ Total Cost | | | | | $c^\star = 5.0$ Total Cost | | | | | $c^\star = 100.0$ Total Cost | | | | | $c^\star = 1000.0$ Total Cost | | | | |
|---|---|---|---|---|---|---|---|---|---|---|---|---|---|---|---|---|---|---|---|---|---|
| | | 1000 | 2000 | 5000 | 10000 | 15000 | 1000 | 2000 | 5000 | 10000 | 15000 | 1000 | 2000 | 5000 | 10000 | 15000 | 1000 | 2000 | 5000 | 10000 | 15000 |
| SURFREE | VA | 10.7 | 7.1 | 4.2 | 3.0 | 2.3 | 16.6 | 11.0 | 6.8 | 4.4 | 3.6 | **57.6** | **57.1** | 33.3 | 20.7 | 18.6 | 60.0 | 56.0 | 56.8 | **56.8** | **57.6** |
| | VA+AS (A-SurFree) | **8.9** | **6.0** | **3.7** | **2.3** | **2.0** | **14.4** | **8.3** | **4.6** | **3.2** | **2.6** | 58.2 | 58.2 | 27.7 | **15.0** | **10.4** | **58.0** | 54.9 | 54.6 | 56.9 | 58.3 |
| HSJA | VA | 37.5 | 29.9 | 17.8 | 10.8 | 7.1 | 49.6 | 40.0 | 26.0 | 18.4 | 13.9 | 57.3 | **56.0** | 55.1 | 52.5 | 49.7 | 56.7 | 58.9 | 59.2 | 57.2 | 58.0 |
| | VA+AS | 38.1 | 28.6 | 18.5 | 9.6 | 6.8 | 47.8 | 38.5 | 23.9 | 16.5 | 11.0 | 57.4 | 56.6 | 55.0 | 47.4 | 44.5 | 56.6 | 60.9 | 57.9 | 59.2 | **57.4** |
| | VA+AGREST | **14.5** | 8.0 | 4.0 | **2.3** | **1.7** | 29.4 | 14.6 | **6.0** | 3.8 | 2.7 | **56.7** | 56.9 | 59.6 | **30.5** | 31.6 | 55.9 | 55.3 | 58.2 | 57.4 | 59.0 |
| | VA+AS+AGREST (A-HSJA) | 15.1 | **7.7** | **3.9** | **2.3** | 1.8 | **27.1** | **14.5** | 6.4 | **3.6** | **2.6** | 58.1 | 58.1 | **40.5** | 31.3 | **21.0** | 58.2 | 56.7 | 57.5 | 56.7 | 58.7 |
| GEODA | VA | 12.5 | 7.3 | 3.1 | 2.2 | 1.8 | 19.7 | 13.7 | 6.5 | 3.3 | 2.5 | 56.8 | 57.7 | 32.5 | 25.5 | 22.6 | 58.2 | **56.5** | 60.3 | **57.4** | **55.8** |
| | VA+AS | 14.4 | 7.1 | 3.2 | 2.0 | 1.9 | 19.9 | 12.6 | 5.5 | 3.3 | 2.5 | 60.2 | 60.6 | 30.6 | 23.1 | 17.6 | **58.0** | 57.0 | **58.0** | 59.3 | 57.3 |
| | VA+AGREST | 8.4 | 4.0 | **2.1** | **1.5** | 1.4 | **13.5** | **8.1** | 3.8 | 2.6 | **2.1** | 58.3 | **55.8** | 35.5 | 19.4 | 17.0 | 58.8 | 59.0 | 58.4 | 57.6 | 57.4 |
| | VA+AS+AGREST (A-GeoDA) | **7.9** | **3.8** | 2.1 | 1.6 | **1.4** | 13.7 | 8.5 | 4.1 | **2.5** | 2.2 | **55.9** | 58.5 | 27.5 | **18.1** | **15.2** | 59.2 | 60.6 | 58.4 | 57.7 | 58.0 |
| CGBA | VA | 11.3 | 6.0 | 2.3 | 1.3 | 1.0 | 16.6 | 11.2 | 4.6 | 2.4 | 1.6 | 57.1 | 54.9 | 32.4 | 23.7 | 18.6 | **56.1** | 56.7 | 59.5 | 61.3 | 59.2 |
| | VA+AS | 11.2 | 5.6 | 2.4 | 1.3 | 1.0 | 16.4 | 9.5 | 4.3 | 2.2 | **1.5** | 56.2 | 56.6 | 28.1 | 17.2 | 14.8 | 59.0 | 58.0 | 59.9 | 58.3 | 55.5 |
| | VA+AGREST | **6.5** | **3.4** | 1.8 | 1.2 | **0.9** | 14.0 | 7.9 | 3.8 | 2.2 | 1.8 | **55.0** | 55.7 | 35.0 | 19.5 | 14.1 | 56.3 | **55.4** | 59.3 | 57.6 | **53.5** |
| | VA+AS+AGREST (A-CGBA) | 7.6 | 3.5 | **1.7** | 1.2 | 1.0 | **12.2** | **7.5** | **3.4** | 2.2 | 1.9 | 58.1 | 60.5 | **21.5** | **16.5** | **12.2** | 58.2 | 58.2 | **55.9** | 58.8 | 57.2 |

# G  ASYMMETRIC SEARCH (AS) ILLUSTRATION

Fig. 7 provides a visual example of the Asymmetric Search (AS) algorithm running with parameters $\tau = 0.1$ and $c^\star = 2$. The illustration shows the iterative progression and query evaluations leading to successful convergence near the decision boundary.

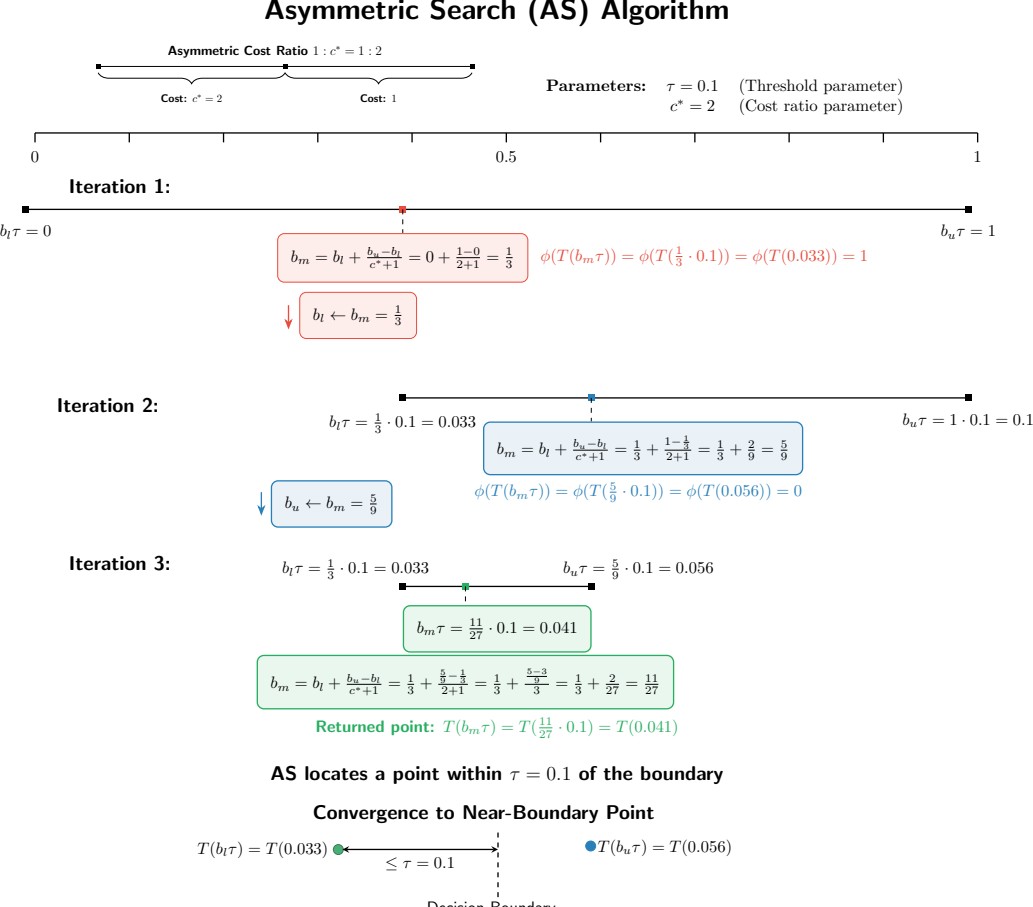

Figure 7: Asymmetric Search (AS) illustration.

## H    ASYMMETRIC ATTACKS AGAINST CLIP

We evaluate the robustness of vision-language models (VLMs), such as CLIP (Radford et al., 2021), against **stealthy** adversarial attacks. Our experiments cover both the zero-shot and fine-tuned versions of CLIP. We apply our asymmetric attacks to these models and observe substantial improvements over stealthy baselines. As shown in Fig. 8, after making 300 total queries, asymmetric methods achieve 40–60% lower $\ell_2$ distortion compared to Stealthy HSJA.

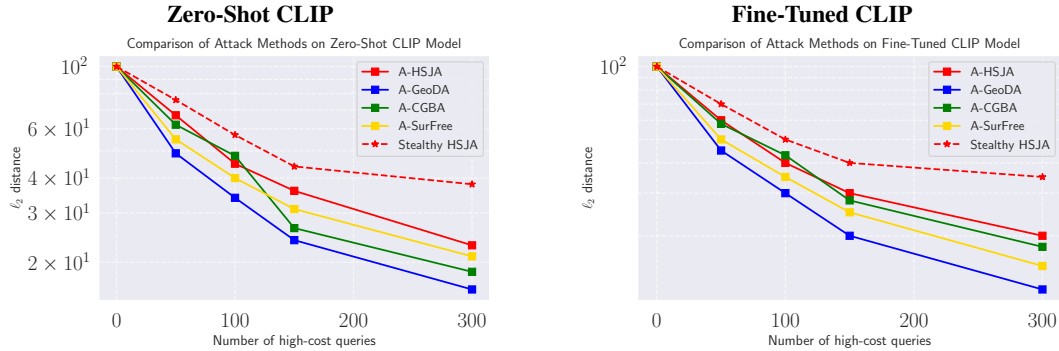

Figure 8: Performance of various asymmetric attacks compared to Stealthy HSJA on CLIP.

## I    CONCEPTUAL ILLUSTRATION

In this section, we show the conceptual illustration of the vanilla gradient estimation and our proposed gradient estimation AGREST.

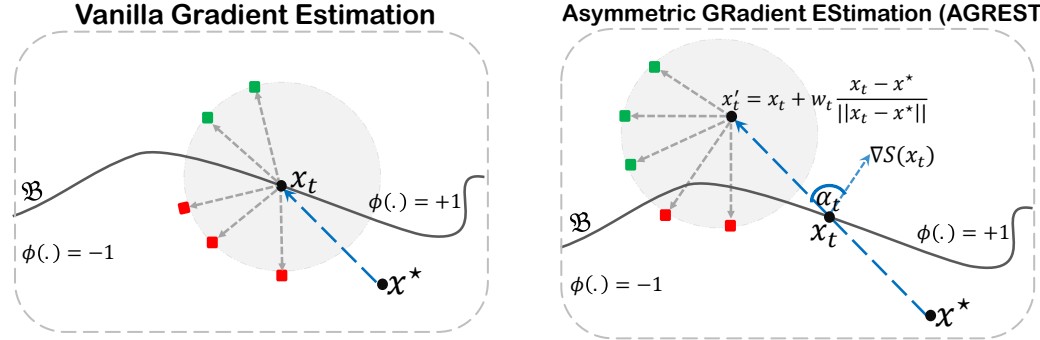

Figure 9: **Comparison of vanilla gradient estimation and its asymmetric counterpart.** Vanilla sampling results in roughly half high-cost and half low-cost queries, whereas AGREST reduces the frequency of high-cost queries by shifting the sampling region and weighting outcomes accordingly.

## J    NOTE ON THE USE OF LARGE LANGUAGE MODELS

Large language models (LLMs) were utilized exclusively for the purpose of writing and refining the manuscript. LLMs were employed to enhance grammar, increase clarity, and rephrase sentences for improved readability. All research concepts, experiments, and analyses were carried out without the assistance of LLMs.

