# OpenReview forum: "A General Framework for Black-Box Attacks Under Cost Asymmetry"
_ICLR.cc/2026/Conference — ICLR 2026 Poster_

### Official Review · Reviewer_gn6L · 2025-10-29

**Soundness:** 3
**Presentation:** 3
**Contribution:** 3
**Rating:** 6
**Confidence:** 3

**Summary:**

This paper introduces two techniques for creating adversarial examples against black-box image classifiers when queries have asymmetric cost. These techniques, AS and AGREST, can be plugged into existing attack frameworks (e.g. HSJA, GeoDA...) to substantially increase their performance under asymmetric query costs. AS is a variant of binary search where the interval is split in two parts of length proportional to the cost of the respective queries, and can also be seen as a better version of multi-stage line search. AGREST estimates the gradient around a boundary point by sampling from a hypersphere that is not exactly at the boundary, but shifted toward the low-cost class in order to make fewer high-cost queries, and adjustments are made to estimate the gradient at the original boundary point.

**Strengths:**

* solid theoretical grounding
* the techniques can be plugged into existing attack frameworks
* introduction of asymmetric gradient estimation, which addresses a question left as an open problem in prior work
* good experimental results
* source code provided

**Weaknesses:**

* introduction of a new hyperparameter (m) which is set by evaluating the performance of the attacks under different values, and presumably would result in degraded performance when attacking an unknown classifier for which the value of m may not be optimal.
* no tests on binary classifiers, only on multi-class classifiers

**Questions:**

1) In Table 1, are you able to report confidence intervals?

2) Table 1: intuitively, your method must increasingly outperform other methods as c* grows. That's why I find the HSJA results on ViT-B/32 at c* = 2 surprising: 18.3, 18.3, 2.7 and 2.7. I would have expected 4 values instead of just 2, and results more similar to HSJA on ResNet-50 at c* = 2 or c* = 5. In most other cases, the ResNet-50 and ViT-B/32 results are roughly the same. Is the experiment underpowered? What is the sample size? As a sanity check, did you verify that the performance is the same when you set c* = 1?

3) When you mention the median L2 distance, what is the sample size?

4) Table 5: what do you mean by "Higher queries"? What is the value of c*?

5) Figure 1: what is the cost ratio?

6) 107: there's no Fig. 1 (left). I believe this should be Fig. 3 (right).

7) Do your results hold with a binary classifier, consistent with the motivation of the paper (evading a content moderation classifier) and similar to Debenedetti et al. (e.g. ImageNet-Dogs)?

8) Do you take query discretization into account, as recommended for future work by Debenedetti et al.?

9) Does your method retain appropriate performance when the hyperparameter m is estimated on the same attack but a different classifier than the target one, as would be the case in practice?

Suggestions:
* You could consider converting Table 1 into a plot.

Typos:
 * 377: duplicate Chen et al.
  * 244: I believe that what you call a random variable is instead a sample / realization from a random variable.
    Rating 6
    Confidence

---

> ### Author Response · Authors · 2025-11-23
>
> We thank the reviewer for their helpful suggestions and for identifying multiple typos. Below, we address the raised questions and concerns.
>
> 1. To obtain confidence intervals, we ran each HSJA experiment three times. The results are reported in the following table:
>
> | Method         | c* = 2            | c* = 5            | c* = 100          | c* = 1000          |
> |----------------|-------------------|-------------------|-------------------|--------------------|
> | VA             | 2.2967 ± 0.1156   | 2.3167 ± 0.0896   | 2.3200 ± 0.0942   | 2.2300 ± 0.0779    |
> | VA+AS          | 2.9533 ± 0.1087   | 2.7900 ± 0.0557   | 2.6633 ± 0.0260   | 2.4200 ± 0.0471    |
> | VA+AGREST      | 4.6800 ± 0.0216   | 4.0467 ± 0.1462   | 2.5067 ± 0.0170   | 2.2467 ± 0.0618    |
> | VA+AS+AGREST   | 22.7633 ± 0.4179  | 19.7333 ± 0.0741  | 15.6000 ± 0.7101  | 12.6300 ± 0.2900   |
>
> 2. We appreciate the reviewer’s observations regarding the closeness of the AS and vanilla variants. The reason is that most queries generated by HSJA and GeoDA are used for gradient estimation; thus, by Amdahl’s law, improvements to the AS component yield limited overall gains. Consequently, both vanilla and AS variants require nearly the same number of iterations in this setting, leading to very similar median values (often within one decimal place). We also performed the sanity check suggested by the reviewer, and the results remained consistent.
>
> 3. We compute all medians over 500 randomly selected, correctly classified ImageNet images.
>
> 4. The corresponding cost ratios are as follows: when the total query budget is 150,000, the associated $c^\star$ is 100; when the total budget is 250,000, the associated $c^\star$ is 1000. We thank the reviewer for pointing out the potential confusion, and we will clarify this in the final version.
>
> 5. For this figure, $c^\star = 10^4$.
>
> 6. Thank you for pointing out this typo. Here, we are referring to Figure 1, where our method produces perturbations with an $\ell_2$-norm below 10 while requiring far fewer queries than the Stealthy attacks.
>
> 7. We believe that our attack would also apply in the binary-output setup, as the underlying methodology remains the same. Our experiments on the CIFAR-10 dataset, where the number of classes is significantly smaller than in ImageNet, already indicate that the method continues to provide substantial improvements.
>
> **CIFAR-10 on ResNet-20: Results for HSJA**
> Even in this setting, our method achieves up to a 40% reduction in the perturbation norm.
>
> | Variant              | (2, 10000) | (5, 15000) | (100, 150000) | (1000, 250000) |
> |----------------------|------------|------------|----------------|----------------|
> | VA                   | 0.130      | 0.148      | 0.186          | 0.584          |
> | VA + AS              | 0.129      | 0.147      | 0.184          | 0.506          |
> | VA + AGREST          | 0.132      | 0.142      | 0.137          | 0.409          |
> | VA + AGREST + AS     | 0.133      | 0.139      | 0.130          | 0.309          |
>
> Nonetheless, we agree that including experiments for the binary-output case is important, and we will add these experiments in the final version.
>
> 8. Query quantization can indeed improve robustness against black-box attacks. However, attacks optimized for the $\ell_\infty$ norm are typically effective at breaking such defenses. A complete evaluation would therefore require identifying the appropriate parameters for $\ell_\infty$-based attacks. Nonetheless, to assess the current improvements of our $\ell_2$-based attacks under quantization, we report HSJA results for $c^\star = 1000$:
>
> | Method         | c* = 100 | c* = 1000 |
> |----------------|-----------|-----------|
> | VA             | 9.13      | 37.60     |
> | VA+AS          | 8.82      | 36.93     |
> | VA+AGREST      | 6.70      | 34.79     |
> | VA+AS+AGREST   | 7.08      | 31.48     |
>
> 9. Regarding the selection of $m$, while one could potentially obtain better results by identifying the optimal $m$ for each model and each $c^\star$, our experiments show that this may not be necessary. As demonstrated in the ViT results, the value of $m$ optimized for ResNet at $c^\star = 1000$ still provides substantial improvements across both ViT models and all cost ratios. This suggests that the optimal $m$ can transfer across architectures, even when the models are not particularly similar (Table 4).

---

### Official Review · Reviewer_JvF8 · 2025-10-29

**Soundness:** 3
**Presentation:** 3
**Contribution:** 3
**Rating:** 8
**Confidence:** 5

**Summary:**

This paper addresses the practical and important problem of asymmetric query costs in decision-based black-box adversarial attacks, where queries to a model may incur different costs depending on their output. The authors challenge the common assumption of uniform query costs and propose a general framework that can handle arbitrary cost ratios between "low-cost" (e.g., benign) and "high-cost" (e.g., flagged) queries. The core contributions are two novel techniques: Asymmetric Search (AS), a cost-aware alternative to binary search for finding the decision boundary, and Asymmetric Gradient Estimation (AGREST), a method that shifts the gradient sampling distribution towards the low-cost region and uses importance weighting to maintain estimation accuracy. Through extensive experiments on models like ResNet, ViT, and CLIP, the authors demonstrate that their framework significantly outperforms standard attacks and prior "stealthy" attacks, achieving lower perturbation norms for a given total cost budget.

**Strengths:**

1. The work tackles a highly significant problem with direct real-world implications. In many practical scenarios, such as content moderation systems or pay-per-query APIs, the cost of a query is not uniform. By generalizing the problem from the niche "stealthy attack" setting (where benign queries have zero cost) to one with arbitrary cost ratios, the authors dramatically increase the practical relevance and applicability of decision-based attacks.

2. The proposed techniques, AS and AGREST, are both novel and technically sound. While AS is an intuitive and clever modification of a standard procedure, AGREST is a more profound contribution. The idea of shifting the sampling center away from the boundary into the low-cost region and compensating with importance sampling is an elegant solution to the core challenge of balancing cost reduction with gradient estimation quality. This is supported by solid theoretical analysis (Theorems 2 and 3), which adds a layer of rigor and confidence to the method.

3. The experimental evaluation is comprehensive and convincing. The authors validate their framework across multiple modern and diverse architectures (CNNs, Transformers, VLMs), demonstrating its general applicability. The comparisons are thorough, including not only vanilla baselines but also a direct and fair comparison against the most relevant prior work, Stealthy HSJA. The ablation studies (Table 1) are particularly effective, clearly isolating the individual and combined benefits of AS and AGREST. The reported performance gains—reducing perturbation norms by up to 40% in some cases—are substantial, not marginal.

4. The paper is exceptionally well-written. The motivation is clearly established using a tangible example (NSFW detection). The methodology is presented logically, aided by clear notation and insightful illustrations (Figure 2). The paper is easy to follow from problem formulation to experimental results, making the complex ideas accessible to the reader.

**Weaknesses:**

1. The paper's contributions and evaluations are focused exclusively on improving gradient-based decision attacks (HSJA, GeoDA, etc.). However, there exists another class of decision-based attacks that do not rely on explicit gradient estimation, such as evolutionary algorithms[1] or random walks[2]. These methods might be naturally resilient to query-intensive steps and could potentially be adapted to the asymmetric cost setting with simple modifications to their search strategy. The lack of discussion or comparison to this family of attacks makes the evaluation feel slightly incomplete. While gradient-based methods are the state-of-the-art in terms of query efficiency in the symmetric setting, acknowledging and contextualizing these alternative approaches would strengthen the paper's positioning.

2. AGREST introduces a new hyperparameter, m (the overshooting scheduler rate). According to Appendix D.2, this parameter is tuned on a small set of images and then fixed for all experiments, irrespective of the cost ratio c*. This simplifies the method's application but raises concerns about its robustness. The performance might be sensitive to this choice, and an optimal m for c*=100 may not be optimal for c*=10,000.

3. Reliance on the Local Linearity Assumption: The theoretical justification for AGREST relies on the assumption that the decision boundary is locally linear. While the authors acknowledge this is a common assumption in the literature, and the strong empirical results suggest the method is robust in practice, a deeper discussion of the limitations would be beneficial. For instance, on a highly curved boundary, the shifted sampling region of AGREST could lead to a biased gradient that is not fully corrected by the re-weighting, potentially impacting performance more than a standard estimator.

[1] Efficient decision-based black-box adversarial attacks on face recognition. CVPR, 2019.

[2] Aha! Adaptive History-Driven Attack for Decision-Based Black-Box Models. ICCV, 2021.

**Questions:**

1. The paper focuses on enhancing state-of-the-art gradient-based attacks. Could you elaborate on why non-gradient-based methods, such as a modified Boundary Attack or an evolutionary strategy designed to favor low-cost regions, were not considered as baselines? Do you hypothesize that your gradient-based framework would still be superior, and if so, why?

2. Regarding the hyperparameter m: How sensitive is the attack's performance to the choice of this value? Could you provide a brief sensitivity analysis? Furthermore, have you considered an adaptive schedule where m could be adjusted dynamically based on the cost ratio c* or the attack's progress?

3. The framework is presented for non-targeted attacks. What do you foresee as the primary challenges in extending AS and AGREST to the targeted attack setting, where there may be multiple "high-cost" classes (the original class and non-target classes) and one "low-cost" class (the target class)?

---

> ### Author Response · Authors · 2025-11-23
>
> We thank the reviewer for their insightful feedback. Regarding the noted weaknesses, we agree that discussing the limitations of the linearity assumption, particularly in regions with high curvature decision boundaries, is important. We will add a discussion on this assumption and its potential failure cases. We also agree that $m$ should not behave as a brittle hyperparameter. In our method, $m$ affects only the scheduler used to approximate the angle $\alpha_t$ (Alg. 2); it does not determine the low-cost and high-cost trade-off, which is instead governed by $c^\star$ and the optimal overshoot $\omega^\star$ (Thm. 3; Alg. 3, lines 1–2), nor the reweighting mechanism, which relies on the empirical low-cost rate $\hat{p}_t$ (Alg. 3, lines 12–13). This already limits sensitivity to $m$. Empirically, App. D.2 shows a broad performance plateau with respect to $m$, and we use the same $m$ values across all $c^\star$ while still observing consistent improvements (Table 1; Fig. 3; Fig. 7). We now address the specific questions raised by the reviewer:
>
> 1. We appreciate the reviewer for mentioning that non-gradient-based methods also may be useful since they prioritize low-cost regions. Although that’s true, throughout our experiments we observed that these attacks are somehow similar to Stealthy HSJA in terms of low-cost to high-cost ratio. Furthermore, controlling the ratio between high-cost and low-cost queries is challenging in these setups, making the generalization of these methods for a given c* a challenging, yet worthy, goal. However, we do agree that we should mention these methods and add them as baselines in the final version.
>
> 2. To address the reviewer concern, we provide results of running HSJA with AGREST on 25 randomly selected images for different values of c* and m.
>
> | c*    | m = 0.02 | m = 0.04 | m = 0.06 | m = 0.08 | m = 0.10 |
> |-------|----------|----------|----------|----------|----------|
> | 2     | 2.45     | 2.91     | 2.87     | 2.54     | 3.16     |
> | 5     | 3.52     | 3.59     | 3.75     | 3.37     | 3.62     |
> | 100   | 3.02     | 4.36     | 3.43     | 4.20     | 4.95     |
> | 1000  | 17.69    | 19.71    | 23.75    | 22.84    | 30.10    |
> | 10000 | 12.91    | 17.30    | 23.72    | 24.728   | 24.004   |
>
> It is also worth noting that we use the values obtained from Resnet hyper parameter tuning for ViT experiments. This can also suggest transferring m from a known model for a specific c*. Regarding adaptive selection of m, we acknowledge that this can be a great idea. For example, we have tried some heuristic-based adaptive methods based on the number of queries and have seen some improvements, offering an interesting direction for future work. However, to avoid further complication we assume m is fixed in our methodology.
>
> 3. That is actually an interesting question. We believe a useful example that can give us some insight is one-vs-all linear classification. For the scenario, mentioned by the reviewer, intuitively, the center of sampling should be in the low-cost region. However, one main challenge here is to obtain how far we should go from the boundaries of the other two high-cost regions. This brings us to the next question; what direction should we use for the overshooting, which we think would be the main challenge here, since it heavily depends on the geometry of boundaries.

---

### Official Review · Reviewer_mvch · 2025-11-01

**Soundness:** 3
**Presentation:** 2
**Contribution:** 2
**Rating:** 4
**Confidence:** 3

**Summary:**

This paper addresses the practical limitation of decision-based black-box attacks that assume all queries have equal cost. The paper proposes a new general framework for asymmetric black-box attacks to minimize the total query cost when certain queries are more expensive. The framework introduces two core components: Asymmetric Search, a boundary search method that splits the search interval based on the query cost ratio to minimize the expected cost; and Asymmetric Gradient Estimation, which shifts the sampling distribution to favor low-cost queries and applies differential weighting for variance reduction. Empirically, the proposed asymmetric framework consistently achieves lower total query costs and smaller adversarial perturbations than existing methods, including prior stealthy attacks.

**Strengths:**

- The paper tackles the highly practical and often overlooked problem of cost asymmetry in black-box attacks, making the research highly relevant to real-world deployment scenarios.
- The proposed method can be integrated with existing decision-based attacks, offering flexibility.
- The methods consistently and substantially outperform existing methods by achieving a significantly lower total query cost while maintaining the quality of the adversarial examples.

**Weaknesses:**

- The claim that the method is superior to existing black-box attacks is unconvincing to me. More specifically, asymmetric cost in Eqn. 3 is optimized in the proposed method and also used in the evaluation, while other methods are optimized with **different objectives** ($c^*=0$ or $\infty$). Incorporating the new objective seems straightforward. I acknowledge the novelty in the problem setting and theoretical analysis, but the novelty in the method is quite trivial to me.
- The main weakness of this paper is the limited evaluation. The experiments only contain three models (ResNet50 and two ViT variants), one dataset (ImageNet), and **no defense**. Especially, there are many adaptive defenses for black-box attacks, which should be considered to support the claim that the method is effective and reduces attack cost compared to other stealthy attacks. Furthermore, the paper should dedicate a section to discuss possible adaptive strategies to defend against AS and AGREST.
- The experiments report the performance with different values of $c^* $, but the paper does not propose any method or heuristic to choose $c^*$, limiting the practicality.

**Questions:**

Please see Weaknesses.

---

> ### Author Response · Authors · 2025-11-23
>
> We appreciate the reviewer’s feedback and address the weaknesses they have identified as follows:
>
> 1. We appreciate this concern and clarify that our empirical gains are not an artifact of a special objective and that adapting existing attacks to arbitrary ($c^{\star}$) is not straightforward.
>  (1) Evaluation metric. All methods are evaluated under the same cost function in Eq. (4) for a given ($c^{\star}$). Standard decision‑based attacks are implicitly optimized for the symmetric case ($c^{\star}=1$) (all queries equal), while stealthy methods are optimized for the extreme case ($c^{\star}=\infty$) (only high‑cost queries counted). In the moderate‑asymmetry regime ($c^{\star}=2,5$), our asymmetric variants give only modest improvements over vanilla methods (Tab. 1, Tabs. 5–6), which is expected since the vanilla objective closely matches the evaluation metric there. In contrast, in the extreme‑asymmetry regime, we compare directly against Stealthy HSJA, which is explicitly optimized for ($c^{\star}=\infty$). Under the same objective used in Debenedetti et al., counting only high‑cost queries as cost, our attacks achieve strictly lower $\ell_{2}$ distortion for any fixed number of high‑cost queries (Fig. 3, Fig. 7). Thus, our superiority does not come from choosing a metric that disadvantages baselines; it holds even when we adopt the baselines’ own objective.
> (2) Non‑triviality of the algorithmic changes. Stealthy HSJA already represents the “straightforward” adaptation of HSJA to asymmetric costs: replace binary search by an egg‑dropping / line search and abandon HSJA’s Monte Carlo gradient estimator in favor of an OPT‑style estimator, because extending HSJA’s estimator was considered difficult. In contrast, our AGREST estimator keeps the Monte Carlo structure but re-optimizes the sampling distribution and reweighting to maximize expected cosine similarity between the estimated and true gradients under a cost constraint (Eq. (11)). The resulting parameters; overshoot ($\omega_t^{\star}$), reweighting ($\beta_{t}^{\star}=p_{t}(\omega_t^{\star})$), and batch size ($n_{t}^{\star}$), are the solution of a non‑trivial high‑dimensional optimization problem (Thms. 2–3) and are not obtainable by simply plugging the new cost into existing code. Similarly, our Asymmetric Search is not an arbitrary change of split ratio: we derive the $1:(c^{\star})$ rule by minimizing expected cost and prove that it improves upon standard binary search by a factor ($\Theta(\log(c^{\star}+1))$) (Thm. 1), while reducing to classical binary search at ($c^{\star}=1$) and to line search at ($c^{\star}\to\infty$).
>  Overall, the novelty of our work lies in a general framework that: (i) formalizes arbitrary cost asymmetries, (ii) derives principled, cost‑aware replacements for both search and gradient estimation, and (iii) yields attackers that dominate both vanilla and stealthy methods not only under our new objective but also under the original stealthy objective ($c^{\star}=\infty$).

---

> ### Author Response · Authors · 2025-11-23
>
> 2. We agree with the reviewer that the effectiveness of an attack is ultimately determined by its performance against defenses, particularly adaptive ones. It is worth noting that, similar to prior work on developing black-box attacks [1–5], our primary focus was evaluating performance against standard (undefended) models. Nonetheless, we also report the performance of the HSJA variants for \( c^\star = 100 \) and \( c^\star = 1000 \) under a quantization-based defense.
>
> | Method         | c* = 100 | c* = 1000 |
> |----------------|-----------|-----------|
> | VA             | 9.13      | 37.60     |
> | VA+AS          | 8.82      | 36.93     |
> | VA+AGREST      | 6.70      | 34.79     |
> | VA+AS+AGREST   | 7.08      | 31.48     |
>
> We also evaluate the attack in the same setting against the PGD-trained model (\( \ell_2 = 3 \)) from the robustness library [6], which exhibits a roughly 20% drop in clean accuracy.
> | Method         | c* = 100  | c* = 1000 |
> |----------------|-----------|-----------|
> | VA             | 67.43     | 127.18    |
> | VA+AS          | 66.02     | 116.59    |
> | VA+AGREST      | 61.88     | 108.75    |
> | VA+AS+AGREST   | 60.39     | 85.85     |
>
> [1] Jianbo Chen, Michael I Jordan, and Martin J Wainwright. Hopskipjumpattack: A query-efficient decision-based attack. In 2020 ieee symposium on security and privacy (sp), pp. 1277–1294. IEEE, 2020.
> [2] Ali Rahmati, Seyed Mohsen Moosavi-Dezfooli, Pascal Frossard, and Huaiyu Dai. Geoda: a geometric framework for black-box adversarial attacks. In Proceedings of the IEEE/CVF conference on computer vision and pattern recognition, pp. 8446–8455, 2020.
> [3] Minhao Cheng, Simranjit Singh, Patrick Chen, Pin-Yu Chen, Sijia Liu, and Cho-Jui Hsieh. Sign-opt: A query-efficient hard-label adversarial attack. arXiv preprint arXiv:1909.10773, 2019.
> [4] Md Farhamdur Reza, Ali Rahmati, Tianfu Wu, and Huaiyu Dai. Cgba: Curvature-aware geometric black-box attack. In Proceedings of the IEEE/CVF International Conference on Computer Vision, pp. 124–133, 2023.
> [5] Edoardo Debenedetti, Nicholas Carlini, and Florian Tramèr. Evading black-box classifiers without breaking eggs. In 2024 IEEE Conference on Secure and Trustworthy Machine Learning (SaTML), pp. 408–424. IEEE, 2024.
> [6] Logan Engstrom, Andrew Ilyas, Hadi Salman, Shibani Santurkar, and Dimitris Tsipras. Robustness (Python Library). https://github.com/MadryLab/robustness. 2019.

---

> ### Author Response · Authors · 2025-11-23
>
> 3. We thank the reviewer for raising this point. In our formulation, ($c^{\star}$) is not a free algorithmic tuning knob but part of the problem specification: it encodes the relative cost of high‑cost vs low‑cost queries,
>  $$
>  \text{cost} = Q_{\text{non-flagged}} + c^{\star} Q_{\text{flagged}},\quad
>  c^{\star} = \frac{c_{\text{flagged}} + c_0}{c_0},
>  $$
>  where ($c_0$) is the base cost of any query and ($c_{\text{flagged}}$) is the additional penalty for a flagged query (Eq. (4)). This is analogous to choosing a perturbation budget ($\varepsilon$) in white‑box attacks or a cost matrix in cost‑sensitive classification: there is no single universally optimal value, because different applications assign different penalties to “risky” queries.
> In practice, ($c^{\star}$) can be set from simple domain information. For example, in the NSFW setting of Section 2, suppose each account can send at most ($N$) total posts and at most ($B$) flagged posts before being banned. If the cost of losing an account is 1, then a non‑flagged query consumes ($1/N$) accounts on average, while a flagged query consumes ($1/N + 1/B$). A linear surrogate cost leads to ($c_0 \approx 1/N$) and ($c^{\star} \approx (1/N + 1/B)/(1/N) = 1 + N/B$). With realistic values (e.g. ($N=2400$), ($B=10$) on 𝕏), this yields ($c^{\star} \approx 241$), which lies in the range we study ($10^2$–$10^3$). More generally, practitioners can estimate ($c_0$) from compute/API costs and ($c_{\text{flagged}}$) from manual review effort, ban probability $\times$ ban cost, or other policy constraints, and set ($c^{\star} = 1 + c_{\text{flagged}}/c_0$).
> When the constraint is expressed as a hard cap on flagged queries (e.g. at most ($B_{\max}$) flagged queries per attack), ($c^{\star}$) plays the role of a Lagrange multiplier. In this case, one can perform a small calibration on a handful of images by sweeping ($c^{\star}$) over a logarithmic grid (e.g. $2,5,10^2,10^3$) and choosing the smallest ($c^{\star}$) for which the flagged query count remains below ($B_{\max}$). This is a one‑time, application‑level choice. We will add a short subsection “Choosing ($c^{\star}$) in practice” to clarify these points and include the above NSFW example.
> Finally, we note that our experiments already demonstrate that the proposed framework is robust across a wide range of ($c^{\star}$) values, from nearly symmetric costs ($c^{\star}=2,5$) through moderate asymmetry ($10^2,10^3$) to the extreme stealthy regime ($c^{\star} \to \infty$) (Table 1, Tables 5–6, Fig. 3, Fig. 7). Thus, even if ($c^{*}$) is only approximately specified, our methods still provide meaningful and controllable trade‑offs between total and high‑cost queries.

---

### Official Review · Reviewer_LRJp · 2025-11-02

**Soundness:** 4
**Presentation:** 4
**Contribution:** 2
**Rating:** 6
**Confidence:** 3

**Summary:**

This paper proposes a general framework for decision-based attacks in the setting where flagged queries have a different cost from non-flagged queries. The main difference from prior work is that their framework does not force the attacker to completely rewrite the attack, and instead leverages the same primitives as standard decision-only attacks.

**Strengths:**

- The paper introduces general abstractions for black-box attacks that might be useful to apply in different ways, and provide a clean conceptual model for thinking about these attacks
- The paper is easy to read, and the experiments are thorough
- The theoretical analysis is convincing and not completely unrealistic, actually shedding some light on the design of the estimators.

**Weaknesses:**

- Given that the motivation is to increase the practicality of black-box attacks, it would be interesting to see a real-world study of an adversarial attack pulled off using this method.
- In terms of the actual benefit over Debenedetti et al., it seems like the main thing is just the conceptual advantage of keeping the same primitives as normal black-box attacks, rather than an actual practical advantage
- Conceptually, the AS algorithm seems to be just a weighted binary search, so I'm similarly unsure of the actual novelty there.

Overall, the paper is a conceptually interesting contribution, but I'm not sure of the practical significance, especially without a real-world case study.

**Questions:**

- Is this attack more effective against models that have been defended, either via adversarial training or via gradient masking-style defenses?
- What do the distortions look like qualitatively? I couldn't find any examples in the paper.

---

> ### Author Response · Authors · 2025-11-23
>
> We appreciate the reviewer’s valuable feedback.
>
> Addressing weaknesses:
> We respectfully disagree and will clarify both the practical gains and the technical novelty. Practical advantage. We evaluate under the stealthy objective itself (count only high-cost queries) and observe that our asymmetric variants strictly dominate Stealthy-HSJA across budgets on ResNet-50 (Fig. 3) and CLIP (Fig. 7), e.g., after 300 high-cost queries, our $\ell_2$ is $40-60 \%$ smaller on CLIP. Moreover, when total cost matters, as it does under rate limits, our methods reduce the cheap query explosion by orders of magnitude at comparable flagged counts: in Fig. 1, to reach $\ell_2=10$, Stealthy-HSJA needs 636 flagged / 10M total, whereas A-GeoDA achieves 205 flagged / 210K total and A-HSJA 610 / 710K. These are concrete practical gains, not just conceptual preferences. We attribute them to AGREST, which re-optimizes Monte-Carlo gradient estimation for asymmetric costs; at $c^\star = 10^2, 10^3$ this alone reduces $\ell_2$ by $\approx 40 \%$ vs vanilla (Table 1), while Stealthy-HSJA relies on a less efficient OPT-style estimator.
>
> AS novelty. AS is not an ad-hoc "weighted" tweak. It implements the cost-minimizing split under Assumption A1 and yields a provable expected-cost bound $\mathcal{O}\left(c^{\star} \log (1 / \tau) / \log \left(c^{\star}+1\right)\right)$, a $\Theta\left(\log \left(c^{\star}+1\right)\right)$ improvement over ordinary binary search in the asymmetric model (App. B.1). Empirically, AS reduces search-phase cost by $\approx 2.5 \times$ at $c^{\star}=10^3$ (Fig. 4, left), while recovering binary search at $c^{\star}=1$ and line search as $c^{\star} \rightarrow \infty$. We will correct the main-text statement of Thm. 1 to match the proven bound and add a short ablation contrasting AS with non-optimal fixed-ratio splits to make this distinction explicit.
>
> Addressing questions:
> 1. For the robust model, we used the PGD-trained model with \( \ell_2 = 3 \) (whose clean accuracy drops by roughly 20%) from the robustness library [1]. We note that, due to time constraints, we used the default hyperparameters as in our standard attack; tuning these hyperparameters may further improve attack performance. (HSJA variants)
>
> | Method         | c* = 100  | c* = 1000 |
> |----------------|-----------|-----------|
> | VA             | 67.43     | 127.18    |
> | VA+AS          | 66.02     | 116.59    |
> | VA+AGREST      | 61.88     | 108.75    |
> | VA+AS+AGREST   | 60.39     | 85.85     |
>
> In addition to the white-box defense, we also evaluate quantization as a defense against black-box attacks, which results in a much smaller decrease in clean accuracy compared to white-box defenses. (HSJA variants)
>
> | Method         | c* = 100 | c* = 1000 |
> |----------------|-----------|-----------|
> | VA             | 9.13      | 37.60     |
> | VA+AS          | 8.82      | 36.93     |
> | VA+AGREST      | 6.70      | 34.79     |
> | VA+AS+AGREST   | 7.08      | 31.48     |
>
>
> 2. Qualitative results for the different HSJA variants at \( c^\star = 1000 \) are available at the following link: https://gofile.io/d/loeMyT. We will add more examples to the final version.
>
> [1] Logan Engstrom, Andrew Ilyas, Hadi Salman, Shibani Santurkar, and Dimitris Tsipras. Robustness (Python Library). https://github.com/MadryLab/robustness. 2019.

---

### Meta-Review · Area_Chair_u9Ks · 2026-01-02

**Summary:**

The paper proposes a new black-box attack method in the setting where flagged queries have a different cost from non-flagged queries. The main difference from prior work is that the method does not force the attacker to completely rewrite the attack, and instead leverages the same primitives as standard decision-only attacks. Empirically, the proposed asymmetric framework consistently achieves lower total query costs and smaller adversarial perturbations than existing methods. Although several limitations exist, this is a solid contribution to AI security.

**Reviewer Concerns:**

1. Did not give real-world applications that need this kind of adversarial attack.
2. The experiments were not thorough, in particular have limited consideration against defenses.
3. AGREST introduces a new hyperparameter m, which may result in degraded performance when m is not optimal.

**Reviewer Scores:**

LRJp would keep the score.
mvch would increase the score, but not much.
JvF8 would keep the score
gn6L would keep the score

---

### Decision · Program_Chairs · 2026-01-26

Accept (Poster)